# GENERATIVE PARAMETER-EFFICIENT FINE-TUNING

## ABSTRACT

Fine-tuning pretrained (large) Transformer backbones efficiently for downstream tasks has been extensively explored using both Parameter-Efficient Fine-Tuning (PEFT) methods, such as Low-Rank Adaptation (LoRA) and its variants, as well as more recent Representation-Efficient Fine-Tuning (ReFT) approaches. In both of these formulations, fine-tuning weights for selected pretrained layers are treated as model parameters that are directly learned from the downstream task data, often making them layer-specific. While these methods simultaneously aim for memory efficiency, some approaches, such as VeRA (Vector-based Random matrix Adaptation), may not achieve this consistently in practice. In this paper, we propose a novel approach for generating fine-tuning weights through a configurable layer-sharing mechanism, termed **Generative parameter-effIcient Fine-Tuning (GIFT)**. GIFT uses a simple parameterization scheme involving two linear layers (without bias terms) to enable efficient fine-tuning. This method bridges the gap between PEFT and ReFT, ensuring both parameter and memory efficiency. GIFT can be viewed as a variant of LoRA with parameters shared across layers, conditioned on the pretrained weights, with significantly fewer trainable parameters. Through extensive experiments, we demonstrate that our GIFT consistently achieves superior performance and parameter efficiency compared to baselines on commonsense and arithmetic reasoning tasks, instruction tuning with the Llama family of models, and visual recognition benchmarks with Vision Transformers. Notably, GIFT achieves a 5.7% absolute increase in average accuracy with a 14x reduction in trainable parameters compared to LoRA on the Commonsense170k dataset using `Llama-3 (8B)`, and a 5.4% increase in win rate with a 4x reduction in parameters using `Llama-2 (7B)` during instruction tuning. Our method also attains a slightly higher win rate for instruction tuning than `GPT-3.5 (Turbo 1106)`.

## 1 INTRODUCTION

Fine-tuning pretrained deep neural networks (DNNs) as feature backbones for downstream tasks has been an important and challenging research topic. In recent years, large feature backbones with open weights such as Llama (Touvron et al., 2023a;b; AI@Meta, 2024), termed foundation models (Bommasani et al., 2021), have become ubiquitous. Training such models from scratch is impossible with limited resources, and fine-tuning them entirely may also be costly. This raises questions about which parts of a pretrained model to fine-tune (often as a hyperparameter), and how they should be trained (entailing rigorous formulations).

Efficient fine-tuning in terms of parameters, compute and memory/storage has been extensively explored using both Parameter-Efficient Fine-Tuning (PEFT) methods, such as Low-Rank Adaptation (LoRA) (Hu et al., 2022) and its variants (Zhang et al., 2023b; Dettmers et al., 2023; Lialin et al., 2023; Jie & Deng, 2023; Kopiczko et al., 2023; Gao et al., 2024; Liu et al., 2024), as well as more recent Representation Fine-Tuning (ReFT) (Wu et al., 2024b) approaches. LoRA learns weight-residuals in the low-rank form (i.e., $A^l$ and $B^l$ in Fig. 1 (a)). Motivated from a causal intervention mechanism (Geiger et al., 2024), ReFT steers the pretrained model towards a task by editing the representations of a few selected tokens in a low-dimensional subspace, showing strong performance at a lower parameter cost compared to LoRA, albeit at a slight increase in inference cost as the learnable parameters cannot be merged into the pretrained model like LoRA.

In both, LoRA-based and ReFT-based formulations, fine-tuning parameters for selected pretrained layers are treated as model parameters that are directly learned from the downstream task data, often

Figure 1: Comparisons between (a) LoRA (Hu et al., 2022) and (b) our proposed GIFT.

making them layer-specific. ReFT further entails token selection (i.e., selective token positions to intervene regardless of the sequence), and is applicable to the residual stream (which is weightless and not fine-tuned in PEFT methods). Moreover, while LoRA-based methods simultaneously aim for memory efficiency, some approaches like VeRA (Vector-based Random matrix Adaptation) (Kopiczko et al., 2023), may not achieve this in practice without sacrificing performance on the downstream task. **We are motivated to develop efficient fine-tuning methods where the learnable parameters are not layer or token specific, while ensuring memory efficiency:** *(i) Enabling configurable layer-sharing in learning fine-tuned weights* will result in more efficient and potentially more effective fine-tuning. *(ii) Enabling token-agnosticity will facilitate the exchangability between PEFT and ReFT*, leading to potentially better understanding of PEFT in terms of the relationship between frozen pretrained models and their fine-tuned models for a downstream task.

Specifically, let $W^l \in \mathbb{R}^{d_{out} \times d_{in}}$ denote the pretrained weights of a layer $l \in L$ of a model to be finetuned, and $\hat{W}^l \in \mathbb{R}^{d_{out} \times d_{in}}$ denote the finetuned weights. LoRA learns $\hat{W}^l$ by,

$$\textbf{LoRA:} \quad \hat{W}^l_{d_{out} \times d_{in}} = W^l_{d_{out} \times d_{in}} + B^l_{d_{out} \times r} \cdot A^l_{r \times d_{in}}, \quad (1)$$

where $r$ is the (low) rank ($r \ll \min(d_{in}, d_{out})$). Tied LoRA (Renduchintala et al., 2024) propose to share the residual weights across layers selected for fine-tuning (i.e., $\Delta W = B \cdot A \ \forall l \in L$) to enable layer agnosticity. However, in our ablation studies (Section 4.1), we show that this strategy leads to subpar performance.

The variants of LoRA focus on different parametrization scheme of $B^l \cdot A^l$ by exploiting different constraints in addition to be low-rank. For example, VeRA (Kopiczko et al., 2023) uses *fixed* random matrices for $\mathbf{B}^l$ and $\mathbf{A}^l$ and learns learns $\hat{W}^l$ by,

$$\textbf{VeRA:} \quad \hat{W}^l_{d_{out} \times d_{in}} = W^l_{d_{out} \times d_{in}} + \Lambda^l_{d_{out} \times d_{dout}} \cdot \mathbf{B}^l_{d_{out} \times r} \cdot \Gamma^l_{r \times r} \cdot \mathbf{A}^l_{r \times d_{in}}, \quad (2)$$

where $\Lambda^l_{d_{out} \times d_{dout}}$ and $\Gamma^l_{r \times r}$ are diagonal matrices. Although VeRA can significantly reduce the number of learnable parameters, the rank $r$ needs to be sufficiently high for achieving good performance, which leads to a significant increase in memory consumption and training time in practice (as observed in our experiments).

For ReFT, let $y^l \in \mathbb{R}^{d_{out} \times 1}$ be the activation output (i.e., representation) for a token selected to intervene in the $l$-th layer, DiReFT (Wu et al., 2024b) edits the representation by,

$$\textbf{DiReFT:} \quad \hat{y}^l_{d_{out} \times 1} = y^l_{d_{out} \times 1} + B^l_{d_{out} \times r} \cdot (W^l_{r \times d_{out}} \cdot y^l_{d_{out} \times 1} + b^l_{r \times 1}), \quad (3)$$

which can be viewed as LoRA applied directly to hidden representations at selected intervened positions. DiReFT builds an explicit and simple learnable affine relationship between the edited / fine-tuned representation (the 2nd term) and the representation of the pretrained model $y^l$.

**Our Contributions:** (i) As shown in Fig. 1 (b), we propose a novel approach for generating fine-tuning weights through a configurable layer-sharing mechanism, termed **G̲enerative parameter-effI̲cient F̲ine-T̲uning (GIFT)**. We have,

$$\textbf{Our GIFT:} \quad \hat{W}^l_{d_{out} \times d_{in}} = W^l_{d_{out} \times d_{in}} + \mathcal{G}(W^l_{d_{out} \times d_{in}}; \Theta), \quad (4)$$

$$= W^l_{d_{out} \times d_{in}} + W^l_{d_{out} \times d_{in}} \cdot \phi_{d_{in} \times r} \cdot \psi_{r \times d_{in}}, \quad (5)$$

where in Eqn. 4, $\mathcal{G}(\cdot; \Theta)$ is a weight-generator, which learns to generate the fine-tuning weights directly from the pretrained weights, and $\Theta$ collects parameters of the weight-generator, which are shared by multiple layers (e.g., all the Query layers of a pretrained Transformer model). Eqn. 5 presents a simple and linear parametrization scheme for the weight-generator, $\Theta = (\phi, \psi)$.

(ii) We show that our two-linear-layer parametrized GIFT (Eqn. 5) bridges the gap between PEFT such as LoRA (Eqn. 1) and DiReFT (Eqn. 3), extending the direct and simple relationship between

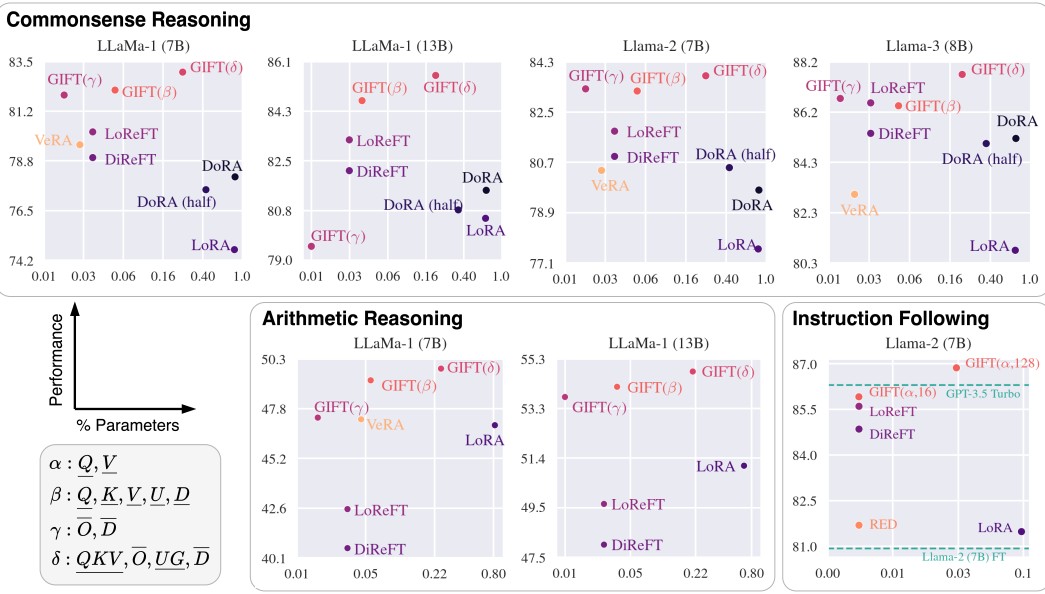

Figure 2: Comparisons of performance vs. trainable parameters between GIFT and baseline methods on three tasks using the Llama model family. All GIFT variants consistently achieve comparable or better performance than prior PEFT (Liu et al., 2024; Kopiczko et al., 2023) and ReFT (Wu et al., 2024b) methods at a much lower parameter cost. See Section 3 for experimental details.

edited representation and pretrained counterparts in the representation space to parameter space in an exchangeable way. Similar to how DiReFT can be viewed as a customized LoRA, our GIFT can be thought of as a variant of LoRA with layer-sharing, conditioned on pretrained weights. GIFT contains significantly fewer trainable parameters, while ensuring both parameter and memory efficiency, and shows superior performance consistently across an extensive series of experiments (see Fig. 2).

## 2 APPROACH

### 2.1 OUR PROPOSED GIFT

Denote by $\Omega_{L \times d_{out} \times d_{in}}$ the pretrained weights of $L$ selected layers in fine-tuning (e.g., all the Query layers). Following the common practice towards efficiency, we enforce a low-rank structure for the weight-generator network $\mathcal{G}(\cdot; \Theta)$ in Eqn. 4. We have,

$$\mathcal{G}(\Omega_{L \times d_{out} \times d_{in}}; \Theta) = \text{Linear}\Bigg( g\bigg( \text{Linear}\big(\Omega_{L \times d_{out} \times d_{in}}; \phi\big); \theta \bigg); \psi \Bigg), \qquad (6)$$

where,

- $\text{Linear}(\Omega; \phi)$ projects the input dimension to a lower dimension (or rank) $r$ with learnable weights $\phi \in \mathbb{R}^{d_{in} \times r}$ without bias terms. Denote by $\Omega_1 \in \mathbb{R}^{L \times d_{out} \times r}$ the output of this layer.
- $\text{Linear}(\cdot; \psi)$ is an output dimension-recovery projection with learnable weights $\psi \in \mathbb{R}^{r \times d_{in}}$ and no bias term. It outputs the learned weight-residuals, $\Delta\Omega \in \mathbb{R}^{L \times d_{out} \times d_{in}}$.
- $g(\cdot; \theta)$ is the low-dimensional generator network, which can be realized by any suitable network specifications. We consider the following schema in this paper:
  - *Transformer*: We treat $\Omega_1$ as a batch of $L$ sequences each consisting of $d_{out}$ tokens in $r$-dim space. We then apply a single Transformer block (Vaswani et al., 2017; Dosovitskiy et al., 2021).
  - *MLP-Mixers*: Similar to vanilla Transformers, we apply a single MLP-Mixer (Tolstikhin et al., 2021) block.
  - *Multi-Layer Perceptrons (MLPs)*: e.g., $g(\Omega_1; \theta) = \text{Linear}(\text{GELU}(\text{Linear}(\Omega_1; \theta_1)); \theta_2)$, where $\theta_1 \in \mathbb{R}^{m \cdot r \times r + m \cdot r}$ and $\theta_2 \in \mathbb{R}^{r \times m \cdot r + r}$ consist of weights and bias terms of the two linear layers with the MLP latent dimension ratio $m$ (e.g., $m = 2$).
  - *Element-wise non-linearity functions* without learnable parameters (i.e., $\theta = \emptyset$): e.g., $g(\Omega_1) = \text{Sigmoid}(\Omega_1)$ or $g(\Omega_1) = \text{GELU}(\Omega_1)$.
  - **The identity operation:** $g(\Omega_1; \theta) = \text{Identity}(\Omega_1) = \Omega_1$ with no learnable parameters $\theta = \emptyset$, which leads to the simple two-layer linear parameterization of GIFT (Eqn. 5).

Through ablation studies (Sec. 4), we show that the two-layer linear parameterization of GIFT is surprisingly effective, and thus our focus in this paper [1]. We rewrite Eqn. 5 here,

$$\hat{W}^l_{d_{out} \times d_{in}} = W^l_{d_{out} \times d_{in}} + W^l_{d_{out} \times d_{in}} \cdot \phi_{d_{in} \times r} \cdot \psi_{r \times d_{in}},$$

$$= W^l_{d_{out} \times d_{in}} \cdot \left( \mathbb{I} + \phi \cdot \psi \right) \triangleq W^l_{d_{out} \times d_{in}} \cdot \Theta_{d_{in} \times d_{in}}, \tag{7}$$

where $\mathbb{I}$ is the identity matrix. The two-linear-layer GIFT can be viewed as a layer-sharing and pretrained weights conditioned variant of LoRA, where we have the counterpart of the layer-specific $B^l_{d_{out} \times r}$ in LoRA, $B^l_{d_{out} \times r} = W^l_{d_{out} \times d_{in}} \cdot \phi_{d_{in} \times r}$, is computed, rather than being treated as direct learnable parameters, by conditioning on the layer-specific pretrained weights and modulating with a layer-agnostic $\phi_{d_{in} \times r}$, and the counterpart of the layer-specific $A^l_{r \times d_{in}}$ in LoRA, $A^l_{r \times d_{in}} = \psi_{r \times d_{in}}(\forall l)$ is directly relaxed to be layer-agnostic.

It is important to note that the "GIFTed" weights (Eqn. 7) are still layer-specific even though the parameters $\phi$ and $\psi$ are shared. This is different from Tied LoRA (Renduchintala et al., 2024), where the residuals are the same across all the layers. In Section 4.1 we show that our GIFT formulation leads to much better performance than simply sharing the weight residuals across layers, which shows the importance of learning layer-specific fine-tuned weights as done in vanilla LoRA and GIFT.

**GIFT can be applied along the $d_{out}$ dimension too.** It is straightforward to learn GIFT along the $d_{out}$ dimension by,

$$(\hat{W}^l_{d_{out} \times d_{in}})^\top = (W^l_{d_{out} \times d_{in}})^\top + (W^l_{d_{out} \times d_{in}})^\top \cdot \phi_{d_{out} \times r} \cdot \psi_{r \times d_{out}}. \tag{8}$$

We will henceforth focus our description of GIFT only on the $d_{in}$ dimension for simplicity.

## 2.2 GIFT BRIDGES PEFT AND REFT

Consider a linear layer with pretrained weights $W^l \in \mathbb{R}^{d_{out} \times d_{in}}$ and the bias term $b^l_{d_{out}}$. The representation/activation for its input $x^l_{N \times d_{in}}$ is $y^l_{N \times d_{out}} = x^l_{N \times d_{in}} \cdot (W^l_{d_{out} \times d_{in}})^\top + b^l_{d_{out}}$. With the GIFT weights $\hat{W}^l_{d_{out} \times d_{in}}$ (Eqn. 7), we have,

$$\hat{y}^l_{N \times d_{out}} = x^l_{N \times d_{in}} \cdot (\hat{W}^l_{d_{out} \times d_{in}})^\top + b^l_{d_{out}} = \hat{x}^l_{N \times d_{in}} \cdot (W^l_{d_{out} \times d_{in}})^\top + b^l_{d_{out}},$$

where $\hat{x}^l_{N \times d_{in}} = x^l_{N \times d_{in}} + x^l_{N \times d_{in}} \cdot (\phi_{d_{in} \times r} \cdot \psi_{r \times d_{in}})^\top = x^l_{N \times d_{in}} \cdot (\Theta_{d_{in} \times d_{in}})^\top$ is the "GIFTed" input activation, the counterpart of the "GIFTed" weights $\hat{W}^l_{d_{out} \times d_{in}}$ (Eqn. 7). Hence, **our GIFT can be equivalently applied to the input activation, rather than the pretrained weights, to achieve the same fine-tuning effect**, maintaining the memory and compute efficiency of LoRA in implementation. Unlike the ReFT (Wu et al., 2024b) that entails a dedicated search for where the representation interventions should apply at the token level, our GIFT eliminates the need of search, enabling token-agnosticity and providing a conceptual shift from the representation intervention.

## 2.3 GIFT AIMS TO "BALANCE" PRETRAINING AND FINE-TUNING

Pretrained Transformer backbones encode diverse knowledge from large-scale pretraining datasets within their weights. Fine-tuning them for a downstream task aims to incorporate new information from the task-specific training data and utilize the information present in the pretrained weights to the fullest extent. To achieve this, the fine-tuned weights can be directly conditioned on the pretrained weights, such that the new information is learned conditionally from the information in the pretrained weights. While LoRA and it's variants use a residual structure to address this, the residual weights are not directly conditioned on the pretrained weights, but rather learned via back-propagation (chain rule) updates. One of the simplest functions that can achieve this explicit conditioning is a linear transformation of the pretrained weights, as leveraged in Eqn. 7. Hence, the fine-tuned weights can also be expressed in the space of the pretrained weights $W_{d_{out} \times d_{in}}$ via $W_{d_{out} \times d_{in}} \cdot \Theta_{d_{in} \times d_{in}}$.

When pretrained Transformer backbones are sufficiently expressive, as is typically assumed in efficient fine-tuning, simpler parameterization methods like GIFT should be more generalizable and better under the principle of Occam's razor. Our ablation studies in Section 4.2 show the effectiveness of the linear parametrization over other schemes.

---

[1]We note that the ablation study is done on the computer vision tasks. So the choice is preliminary. When computing resources are available, we will conduct more thorough ablation studies on language tasks.

## 3 EXPERIMENTS

We conduct extensive experiments across Natural Language Generation, Natural Language Understanding, and Visual Recognition and compare our two-linear-layer parameterized GIFT with various other PEFT methods and ReFT. We also conduct ablation studies on the different parameterization schemes of GIFT. **We use the HuggingFace' PEFT code framework. Our source code is provided in the supplementary.** We describe experiments with Natural Language Understanding on the GLUE dataset (Wang et al., 2018) in Appendix A.1.

*Naming Convention and choice of finetuning layers*: We mainly follow the prior works of selecting layers of pretrained backbones to be fine-tuned on different tasks for fair comparisons. Fig. 3 illustrates the naming convention. We index different components in a Transformer block using their initials ($Q, K, V, O, U, G, D$): $Q$uery, $K$ey, $V$alue and $O$utput projection in MHSA, and $U$p projection, $G$ate projection, and $D$own projection in MLP.

E.g., **GIFT**$_{Q,K,V,U,D}^r$ represents that a separate GIFT is applied for all the components, and the GIFT parameters are shared across all the layers of the same component. We use the preceding superscript $^B$ to represent a *block-wise sharing pattern* we test: $^B$**GIFT**$_{\underline{QKV},\overline{O},\underline{UG},\overline{D}}^r$ in which each Transformer block has its own GIFTs, where we share one GIFT each for $\underline{QKV}, \underline{UG}, \overline{O},$ and $\overline{D}$.

e.g., $r = 16, r = 128$

**GIFT**$^r$ layer name initials, with under/overline showing GIFT is applied along $d_{in}$ or $d_{out}$

e.g., $\underline{Q}, \overline{K},$ or $\underline{QKV}$

Figure 3: Naming convention of our GIFT in experiments.

### 3.1 RESULT HIGHLIGHTS

Fig. 2 shows the consistently better results of our GIFT in extensive experiments, which we highlight as follows:

- On **Instruction Following** (Section 3.2), our GIFT can outperform `GPT-3.5 Turbo` using 0.0311% trainable parameters in fine-tuning Llama-2 (7B), which is the only method to do so in our comparisons.
- On **Commonsense Reasoning** (Section 3.3), our GIFT outperforms both the prior art of PEFT and of ReFT consistently using Llama 1-2-3 model family, often by large margin with less trainable parameters used.
- On **Arithmetic Reasoning** (Section 3.4), our GIFT can outperform all the prior PEFT and ReFT approaches. Unlike VeRA, which performs slightly better than LoRA, GIFT maintains the computational efficiency while achieving better performance. VeRA takes about 1.5 days in training, while our GIFT takes about 4 hours.
- The proposed block-wise sharing $^B$**GIFT**$_{\underline{QKV},\overline{O},\underline{UG},\overline{D}}^r$ shows stronger consistency of achieving better results across tasks (Commonsense Reasoning and Arithmetic Reasoning).

### 3.2 INSTRUCTION FOLLOWING

**Setup**: We follow the experimental setup of ReFT (Wu et al., 2024b), in which Alpaca-Eval 1.0 (Li et al., 2023) is used for evaluating performance based on the win rate against the responses generated by the `text-davinci-003` model using GPT 4 as the annotator. We fine-tune **GIFT**$_{Q,V}^r$ with $r = 16$ and $r = 128$ using the Ultrafeedback dataset (Cui et al., 2023). We ensure that we do not hill-climb on the test set in hyper-parameter search (see Appendix C.2).

**Results**: Table 1 shows that given the same parameter budget ($r = 16$), GIFT outperforms prior methods. With an increased budget ($r = 128$), which is still below LoRA, **GIFT can outperform** `GPT-3.5 Turbo`. Examples of raw generations can be found in Appendix D.

Table 1: Results of fine-tuning Llama-2 (Touvron et al., 2023b) (7B) with GIFT for instruction following. Params (%) are calculated as the ratio between the number of trainable parameters and the total number of parameters in the base model. The preceding superscript, if added, indicates the source of results.

| Method | Params (%) | Win Rate |
|---|---|---|
| [Li et al., 2023] GPT-3.5 Turbo 1106 | - | 86.3 |
| [Li et al., 2023] Llama-2 *Chat* (13B) | - | 81.1 |
| [Li et al., 2023] Llama-2 *Chat* (7B) | - | 71.4 |
| [Wu et al., 2024a] Full Finetuning | 1.0 | 80.93 |
| [Wu et al., 2024a] LoRA (Hu et al., 2022) | 0.1245 | 81.48 |
| RED (Wu et al., 2024a) | 0.0039 | 81.69 |
| DiReFT (Wu et al., 2024b) | 0.0039 | 84.85 |
| LoReFT (Wu et al., 2024b) | 0.0039 | 85.60 |
| GIFT$_{Q,V}^{16}$ | 0.0039 | 85.91 |
| GIFT$_{Q,V}^{128}$ | 0.0311 | **86.87** |

(Llama-2 (7B))

Table 2: Results on eight Commonsense Reasoning benchmarks by fine-tuning the pretrained LLaMA-1 (Touvron et al., 2023a) 7B/13B, Llama 2 (Touvron et al., 2023b) 7B and Llama 3 (AI@Meta, 2024) 8B models. The preceding superscript, if added, indicates the source of results. (VeRA is trained by us using the standard HuggingFace implementation).

| | Method | Params (%) | BoolQ | PIQA | SIQA | HellaS. | WinoG. | ARC-e | ARC-c | OBQA | Avg |
|---|---|---|---|---|---|---|---|---|---|---|---|
| **LLaMA-1 (7B)** | [Liu et al., 2024]PrefT | 0.039 | 64.3 | 76.8 | 73.9 | 42.1 | 72.1 | 72.9 | 54.0 | 60.6 | 64.6 |
| | [Liu et al., 2024]AdapterS | 1.953 | 63.0 | 79.2 | 76.3 | 67.9 | 75.7 | 74.5 | 57.1 | 72.4 | 70.8 |
| | [Liu et al., 2024]AdapterP | 3.542 | 67.9 | 76.4 | 78.8 | 69.8 | 78.9 | 73.7 | 57.3 | 75.2 | 72.3 |
| | [Liu et al., 2024]LoRA (Hu et al., 2022) | 0.826 | 68.9 | 80.7 | 77.4 | 78.1 | 78.8 | 77.8 | 61.3 | 74.8 | 74.7 |
| | [Liu et al., 2024]DoRA (Liu et al., 2024) | 0.838 | 68.5 | 82.9 | 79.6 | 84.8 | 80.8 | 81.4 | 65.8 | 81 | 78.1 |
| | [Liu et al., 2024]DoRA (half) (Liu et al., 2024) | 0.427 | 70 | 82.6 | 79.7 | 83.2 | 80.6 | 80.6 | 65.4 | 77.6 | 77.5 |
| | VeRA$^{4096}$ (Kopiczko et al., 2023) | 0.023 | 70.4 | 82.4 | 79.9 | 91.4 | 81.8 | 83.3 | 67.0 | 80.6 | 79.6 |
| | GIFT$^{64}_{Q,K,V,U,D}$ | 0.052 | 72.4 | 83.4 | 80.2 | 93.9 | 83.8 | 85.8 | **73.4** | 84.4 | 82.2 |
| | DiReFT (Wu et al., 2024b) | 0.031 | 69.5 | 83.0 | 79.0 | 92.5 | 80.5 | 82.2 | 68.0 | 77.5 | 79.0 |
| | LoReFT (Wu et al., 2024b) | 0.031 | 69.3 | 84.4 | 80.3 | 93.1 | 84.2 | 83.2 | 68.2 | 78.9 | 80.2 |
| | GIFT$^{64}_{\overline{O},\overline{D}}$ | 0.016 | 71.5 | 83.4 | 81.1 | 93.6 | 83.7 | 86.1 | 72.0 | 83.9 | 81.9 |
| | $^B$GIFT$^{16}_{QKV,\overline{O},UG,\overline{D}}$ | 0.249 | **73.1** | **84.9** | **81.2** | **94.2** | **84.5** | **87.3** | 73.0 | **85.7** | **83.0** |
| **LLaMA-1 (13B)** | [Liu et al., 2024]PrefT | 0.031 | 65.3 | 75.4 | 72.1 | 55.2 | 68.6 | 79.5 | 62.9 | 68.0 | 68.4 |
| | [Liu et al., 2024]AdapterS | 1.586 | 71.8 | 83.0 | 79.2 | 88.1 | 82.4 | 82.5 | 67.3 | 81.8 | 79.5 |
| | [Liu et al., 2024]AdapterP | 2.894 | 72.5 | 84.9 | 79.8 | 92.1 | 84.7 | 84.2 | 71.2 | 82.4 | 81.5 |
| | [Liu et al., 2024]LoRA (Hu et al., 2022) | 0.670 | 72.1 | 83.5 | 80.5 | 90.5 | 83.7 | 82.8 | 68.3 | 82.4 | 80.5 |
| | [Liu et al., 2024]DoRA (Liu et al., 2024) | 0.681 | 72.4 | 84.9 | 81.5 | 92.4 | 84.2 | 84.2 | 69.6 | 82.8 | 81.5 |
| | [Liu et al., 2024]DoRA (half) (Liu et al., 2024) | 0.347 | 72.5 | 85.3 | 79.9 | 90.1 | 83.6 | 80.8 | 69.7 | 83.6 | 80.8 |
| | GIFT$^{64}_{Q,K,V,U,D}$ | 0.034 | 74.3 | 87.3 | 81.8 | 95.3 | 86.5 | 87.4 | 76.2 | **89.0** | 84.7 |
| | DiReFT (Wu et al., 2024b) | 0.025 | 71.3 | 86.1 | 80.8 | 94.6 | 83.6 | 85.5 | 72.9 | 82.7 | 82.2 |
| | LoReFT (Wu et al., 2024b) | 0.025 | 72.1 | 86.3 | 81.8 | 95.1 | **87.2** | 86.2 | 73.7 | 84.2 | 83.3 |
| | GIFT$^{64}_{\overline{O},\overline{D}}$ | 0.010 | 69.1 | 82.3 | 80.4 | 91.9 | 82.2 | 82.3 | 66.9 | 80.6 | 79.5 |
| | $^B$GIFT$^{16}_{QKV,\overline{O},UG,\overline{D}}$ | 0.201 | **74.6** | **87.9** | **82.3** | **95.6** | 87.1 | **90.3** | **77.9** | **89.0** | **85.6** |
| **Llama 2 (7B)** | [Liu et al., 2024]LoRA (Hu et al., 2022) | 0.826 | 69.8 | 79.9 | 79.5 | 83.6 | 82.6 | 79.8 | 64.7 | 81 | 77.6 |
| | [Liu et al., 2024]DoRA (Liu et al., 2024) | 0.838 | 71.8 | 83.7 | 76.0 | 89.1 | 82.6 | 83.7 | 68.2 | 82.4 | 79.7 |
| | [Liu et al., 2024]DoRA (half) (Liu et al., 2024) | 0.427 | 72 | 83.1 | 79.9 | 89.1 | 83 | 84.5 | 71.0 | 81.2 | 80.5 |
| | VeRA$^{4096}$ (Kopiczko et al., 2023) | 0.023 | 71.9 | 82.2 | 80.0 | 92.2 | 83.3 | 84.3 | 68.8 | 80.5 | 80.4 |
| | GIFT$^{64}_{Q,K,V,U,D}$ | 0.052 | 73.1 | **85.4** | 81.0 | 94.5 | 85.2 | 87.7 | 74.5 | **84.5** | 83.2 |
| | DiReFT (Wu et al., 2024b) | 0.031 | 70.8 | 83.6 | 80.2 | 93.6 | 82.1 | 84.8 | 70.4 | 81.5 | 80.9 |
| | LoReFT (Wu et al., 2024b) | 0.031 | 71.1 | 83.8 | 80.8 | 94.3 | 84.5 | 85.6 | 72.2 | 82.3 | 81.8 |
| | GIFT$^{64}_{\overline{O},\overline{D}}$ | 0.016 | 73.4 | 85.2 | **81.8** | 94.3 | 85.3 | 87.7 | 74.9 | 83.8 | 83.3 |
| | $^B$GIFT$^{16}_{QKV,\overline{O},UG,\overline{D}}$ | 0.249 | **74.5** | 85.0 | 81.5 | **94.9** | **85.8** | **88.5** | **75.8** | 84.1 | **83.8** |
| **Llama 3 (8B)** | [Liu et al., 2024]LoRA (Hu et al., 2022) | 0.700 | 70.8 | 85.2 | 79.9 | 91.7 | 84.3 | 84.2 | 71.2 | 79.0 | 80.8 |
| | [Liu et al., 2024]DoRA (Liu et al., 2024) | 0.710 | 74.6 | 89.3 | 79.9 | 95.5 | 85.6 | 90.5 | 80.4 | 85.8 | 85.2 |
| | [Liu et al., 2024]DoRA (half) (Liu et al., 2024) | 0.361 | 74.5 | 88.8 | 80.3 | 95.5 | 84.7 | 90.1 | 79.1 | 87.2 | 85.0 |
| | VeRA$^{4096}$ (Kopiczko et al., 2023) | 0.018 | 71.6 | 85.7 | 80.7 | 93.8 | 85.2 | 87.6 | 75.6 | 84.1 | 83.0 |
| | GIFT$^{64}_{Q,K,V,U,D}$ | 0.049 | 75.3 | 89.0 | 81.6 | 96.2 | 88.4 | 92.3 | 81.9 | 87.3 | 86.5 |
| | DiReFT (Wu et al., 2024b) | 0.026 | 73.4 | 88.7 | 81.0 | 95.6 | 85.5 | 91.8 | 81.8 | 85.4 | 85.4 |
| | LoReFT (Wu et al., 2024b) | 0.026 | 75.1 | 90.2 | 82.0 | 96.3 | 87.4 | 92.4 | 81.6 | 87.5 | 86.6 |
| | GIFT$^{64}_{\overline{O},\overline{D}}$ | 0.013 | 75.7 | 89.9 | 82.5 | 96.4 | 88.7 | 92.5 | 82.3 | 86.3 | 86.8 |
| | $^B$GIFT$^{16}_{QKV,\overline{O},UG,\overline{D}}$ | 0.209 | **75.9** | **90.4** | **82.7** | **96.6** | **90.0** | **93.6** | **83.5** | **88.9** | **87.7** |

## 3.3 COMMONSENSE REASONING

**Data**: We follow Hu et al. (2023) and Wu et al. (2024b) to use a combined training data of eight benchmarks (i.e., Commonsense170k), and evaluate GIFT on their test sets individually. Examples in the Commonsense170k are formulated as multiple choice questions and consists of BoolQ (Clark et al., 2019), PIQA (Bisk et al., 2020), SIQA (Sap et al., 2019), HellaSwag (Zellers et al., 2019), WinoGrande (Sakaguchi et al., 2021), Arc-e and Arc-c (Clark et al., 2018), and OBQA (Mihaylov et al., 2018) datasets.

**Models**: We fine-tune the pretrained LLaMa-1 7B and 13B, Llama-2 7B and Llama-3 8B models using our GIFT. We compare with LoRA and DoRA using **GIFT$^r_{Q,K,V,U,D}$**. We compare with LoReFT and DiReFT (Wu et al., 2024b) using **GIFT$^r_{\overline{O},\overline{D}}$**. Furthermore, we also evaluate $^B$**GIFT$_{QKV,\overline{O},UG,\overline{D}}$**. Experimental details including test-set-exclusive hyperparameter tuning setup are in Appendix C.4.

**Results**: Table 2 shows the comparison results. **All variants of GIFT consistently outperform the baselines while using significantly less parameters.** Notably, our proposed block-wise sharing $^B$GIFT$_{QKV,\overline{O},UG,\overline{D}}$ with 0.206% trainable parameters, outperforms all the prior methods while using 4 times fewer parameters than LoRA. All variants of GIFT outperform VeRA (Kopiczko et al., 2023). *Note that even though VeRA reduces the number of parameters, it needs a large rank for the fixed random weights (here, we use 4096) for improving performance, resulting in increased training time and GPU usage. In our experiments using the same setup, **VeRA takes ∼5 days and 18.42GB***

*GPU memory to complete training with Llama 2, whereas **GIFT takes less than 1 day (∼17 hours) and 17.73GB GPU memory**. Given the large training time, VeRA is prohibitive for large benchmarks and models. Examples of raw generations can be found in Appendix D.*

### 3.4 ARITHMETIC REASONING

**Data**: We follow Hu et al. (2023) and Wu et al. (2024b) to use a combined training set of four arithmatic reasoning datasets (Math10k), and evaluate on their individual test sets. The Math10k benchmarks consists of AqUA (Ling et al., 2017), GSM8k (Cobbe et al., 2021), MAWPS (Koncel-Kedziorski et al., 2016) and SVAMP (Patel et al., 2021). While models are expected to generate a chain-of-thought before the final answer, we only evaluate on the final answer following (Wu et al., 2024b). Experimental details including test-set-exclusive hyperparameter tuning strategy are in Appendix C.3.

Table 3: Comparisons on Arithmetic reasoning benchmarks by fine-tuning the pretrained LLaMA-1 (Touvron et al., 2023a) 7B. The preceding superscript, if added, indicates the source of results.

| | LLaMA-1 (7B) | | | | | LLaMA-1 (13B) | | | | |
|---|---|---|---|---|---|---|---|---|---|---|
| **Method** | **Params (%)** | **AQuA** | **GSM8k** | **MAWPS** | **SVAMP** | **Avg** | **Params (%)** | **AQuA** | **GSM8k** | **MAWPS** | **SVAMP** | **Avg** |
| [Hu et al., 2023]PrefT | 0.039 | 14.2 | 24.4 | 63.4 | 38.1 | 35.0 | 0.031 | 15.7 | 31.1 | 66.8 | 41.4 | 38.8 |
| [Hu et al., 2023]AdapterS | 1.953 | 15.0 | 33.3 | 77.7 | 52.3 | 44.6 | 1.586 | 22.0 | 44.0 | 78.6 | 50.8 | 48.9 |
| [Hu et al., 2023]AdapterP | 3.542 | 18.1 | 35.3 | 82.4 | 49.6 | 46.4 | 2.894 | 20.5 | 43.3 | 81.1 | 55.7 | 50.2 |
| LoRA (Hu et al., 2022) | 0.826 | 18.9 | 37.5 | 79 | 52.1 | 46.9 | 0.67 | 18.5 | 47.5 | 83.6 | 54.6 | 51.1 |
| VeRA$^{12288}$ (Kopiczko et al., 2023) | 0.042 | 21.3 | 34.0 | 82.8 | 50.7 | 47.2 | | | | - | | |
| GIFT$^{64}_{Q,K,V,U,D}$ | 0.052 | 22.1 | 36.4 | 83.6 | 54.8 | 49.2 | 0.034 | 25.1 | **46.6** | 83.6 | **61.7** | 54.2 |
| DiReFT (Wu et al., 2024b) | 0.031 | 21.3 | 24.1 | 74.5 | 42.7 | 40.6 | 0.025 | 20.5 | 35.8 | 80.8 | 54.8 | 48.0 |
| LoReFT (Wu et al., 2024b) | 0.031 | 21.4 | 26.0 | 76.2 | 46.8 | 42.6 | 0.025 | 23.6 | 38.1 | 82.4 | 54.2 | 49.6 |
| GIFT$^{64}_{\overline{O},\overline{D}}$ | **0.016** | **23.0** | 33.6 | 80.0 | 52.6 | 47.3 | 0.010 | 25.6 | 44.9 | 85.2 | 59.6 | 53.8 |
| $^B$GIFT$^{16}_{QKV,\overline{O},UG,\overline{D}}$ | 0.249 | 22.0 | **37.7** | **84.0** | **55.3** | **49.8** | 0.201 | **26.0** | 46.2 | **86.3** | 60.6 | **54.8** |

**Models**: For fair comparisons with prior works, we finetune the LLaMA-1 7B/13B models.

**Results**: Table 3 shows the comparison results. **All variants of GIFT outperform prior methods.** GIFT$^{64}_{Q,\underline{K},\underline{V},\underline{U},\underline{D}}$ achieves much higher average accuracy while using much fewer parameters than LoRA and DoRA. GIFT$^{64}_{\overline{O},\overline{D}}$ achieves higher average accuracy than LoReFT and DiReFT while using half the parameters. $^B$GIFT$^{16}_{QKV,\overline{O},\underline{UG},\overline{D}}$ outperforms all the prior methods while using 4 times less parameters than LoRA. In contrast to commonsense reasoning, we do not observe a large difference between the performance of GIFT$_{Q,\underline{K},\underline{V},\underline{U},\underline{D}}$ and GIFT$_{QKV,\overline{O},\underline{UG},\overline{D}}$. This suggests that while different variants are suited for different tasks, GIFT$_{QKV,\overline{O},\underline{UG},\overline{D}}$ is robust to different tasks. All variants of GIFT outperform VeRA (Kopiczko et al., 2023). *Again, although VeRA reduces the number of parameters, it needs a large intermediate dimension for the fixed random weights (here, we use 12288). In our experiments, VeRA takes ∼1.5 days and 20.65GB GPU memory, whereas GIFT takes ∼4 hours and 17.73GB GPU memory using the same setup.* Examples of raw generations can be found in Appendix D.

### 3.5 VISUAL RECOGNITION

Table 4: Results on the finegrained visual classification (FGVC) tasks. The number of trainable parameters are reported without the classification head (which has the same number of parameters for all the methods). The GPU memory usage is reported via `torch.cuda.max_memory_allocated()` during training with the batch size 32.

| Method | Params (%) ↓ | GPU Mem (G) ↓ | CUBS | Bird | Flower | Dog | Car | *Avg* |
|---|---|---|---|---|---|---|---|---|
| VPT (Jia et al., 2022) | 0.054 | 2.753 | 87.88 | 84.79 | 98.98 | 84.51 | 82.89 | 87.81 |
| BitFit (Zaken et al., 2022) | 0.097 | 2.673 | 87.75 | 84.61 | **99.32** | 85.23 | 84.01 | 88.18 |
| LoRA (Hu et al., 2022) | 0.172 | **2.622** | 88.00 | 84.94 | **99.32** | 85.36 | **85.92** | 88.71 |
| GIFT$^{16}_Q$ | **0.029** | 2.646 | **89.71** | **86.28** | 99.22 | **87.44** | 84.28 | **89.39** |

**Data.** We evaluate our GIFT on two image classification benchmarks: i) The fine-grained visual classification (FGVC) benchmark contains 5 datasets – Caltech-UCSD Birds (200 classes) (Wah et al., 2011), NABirds (555 classes) (Horn et al., 2015), Oxford Flowers (102 classes) (Nilsback & Zisserman, 2008), Stanford Cars (196 classes) (Gebru et al., 2017), and Stanford Dogs (120 classes) (Khosla et al., 2011). ii) Due to space constraints, we show the results for the VTAB-1k benchmark (Zhai et al., 2019) in Appendix A.2.

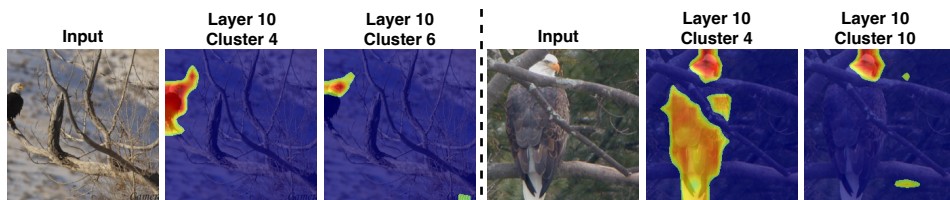

Figure 4: GIFT can play the role of a $r$-way segmentation/token-clustering head that can localize meaningful objects/parts on images. Two examples from NABirds (Horn et al., 2015) benchmark in FGVC are shown here. More examples can be found in Figure 5 in the Appendix.

**Models.** We use the ViT-B/16 architecture (Dosovitskiy et al., 2021) pretrained on ImageNet21k dataset (Deng et al., 2009) using a supervised objective, with the checkpoints from the `timm` package (Wightman, 2019). We apply LoRA and GIFT to the output projection layers in MHSA, which is inspired by observations in (Savadikar et al., 2023) and verified in our ablation studies (Section A.3). All hyperparameters are provided in Appendix C.5.

**Results:** Table 4 and Table 8 (in the appendix) show that our GIFT performs better than other PEFT methods on both FGVC and VTAB-1k, while using fewer parameters. The GPU memory consumption is similar among the different methods with negligible differences. With 5.9 times less parameters used (0.025M vs 0.147M), on FGVC tasks, our GIFT improves LoRA by 0.68% Top-1 accuracy. On VTAB-1k tasks, our GIFT slightly outperforms LoRA by 0.14% Top-1 accuracy.

**Visual Inspection of GIFT**: Let $C^l_{d_{out} \times r} = W^l_{d_{out} \times d_{in}} \cdot \phi_{d_{in} \times r}$ be the transformation using the first linear layer of GIFT, where $W^l_{d_{out} \times d_{in}}$ is the pretrained weights of the output projection layer in MHSA. We show that $C^l_{d_{out} \times r}$ can be used as an emergent segmentation/token-clustering head. Using the fine-tuned model, the activation of the output projection layer is,

$$\hat{y}^l_{N \times d_{out}} = x^l_{N \times d_{in}} \cdot (\hat{W}^l_{d_{out} \times d_{in}})^\top + b^l_{d_{out}}, \tag{9}$$

where $\hat{W}^l$ is the fine-tuned weights (Eqn 5), and $N$ the number of visual tokens in raster order. We compute $r$ heatmaps for visual token clustering by,

$$H^l_{N \times r} = \hat{y}^l_{N \times d_{out}} \cdot C^l_{d_{out} \times r}, \tag{10}$$

which can highlight semantically meaningful parts of the image. We normalize the $r$ heatmaps to $[0, 1]$ individually and use 0.5 as the threshold to generate the visualizations in Figure 4.

## 4 ABLATION STUDIES

### 4.1 SHARING THE WEIGHT RESIDUALS IN LoRA

As mentioned in Section 2, GIFT generates layer-specific weight residuals and fine-tuned weights even though the learnable parameters are shared across layers. We verify that this approach proposed in our GIFT is beneficial over simply sharing the residual weights in LoRA across the layers of the same type, (i.e., $\Delta W^l = B \cdot A \ \forall l \in L$). Table 5 show that this strategy leads to much lower performance than GIFT. This shows that the fine-tuned weights indeed need to be layer specific, and the generative approach in GIFT can achieve this while still maintaining the parameter efficiency of shared weights. This also suggests that methods like (Renduchintala et al., 2024), which impose further restrictions by sharing the residuals across Query, Key and Value components may not scale to more complex datasets.

Table 5: Comparison of Shared LoRA and GIFT on eight commonsense reasoning benchmarks.

| | Method | Params (%) | BoolQ | PIQA | SIQA | HellaS. | WinoG. | ARC-e | ARC-c | OBQA | Avg |
|---|---|---|---|---|---|---|---|---|---|---|---|
| Llama-3 (8B) | Shared LoRA$^{64}_{Q,K,V,U,D}$ | 0.044 | 66.2 | 79.8 | 77.5 | 87.3 | 78.7 | 79.0 | 65.1 | 75.3 | 76.1 |
| | GIFT$^{64}_{Q,K,V,U,D}$ | 0.049 | **75.3** | **89.0** | **81.6** | **96.2** | **88.4** | **92.3** | **81.9** | **87.3** | **86.5** |

### 4.2 DIFFERENT PARAMETERIZATION SCHEMAS FOR GIFT

We evaluate the various schema proposed for GIFT (Section 2) on the FGVC benchmark using the same settings as Section 3.5. As seen from Table 6, the simple two-linear layer formulation achieves better or equivalent performance than all other schema at a lower parameter cost. We hypothesize that when a downstream task is out of distribution to the pretraining non-linear relationships between fine-tuning weight-residuals and pretrained weights could be entailed to be helpful, which we also leave for future investigation.

Table 6: Comparisons between various parameterization schemes of GIFT on the FGVC benchmark.

| Schema | #Params (M) ↓ | GPU Mem (G) ↓ | CUBS | Bird | Flower | Dog | Car | *Avg* |
|--------|---------------|----------------|------|------|--------|-----|-----|-------|
| Identity | 0.025 | 2.65 | **89.71** | **86.28** | 99.22 | **87.44** | 84.28 | **89.39** |
| Sigmoid | 0.025 | 2.65 | 89.56 | 84.61 | 99.20 | 86.69 | 84.04 | 88.82 |
| GeLU | 0.025 | 2.65 | 89.70 | 85.30 | 99.19 | 86.71 | 83.81 | 88.94 |
| MLP | 0.036 | 2.65 | 89.06 | 85.44 | **99.30** | 86.17 | 84.24 | 88.84 |
| Transformer | 0.027 | 2.65 | 89.56 | 86.23 | 99.24 | 86.31 | 84.26 | 89.12 |
| MLP Mixer | 0.125 | 2.65 | 88.76 | 86.21 | 99.25 | 86.35 | **85.66** | 89.25 |

## 5 RELATED WORK

**Parameter Efficient Fine-tuning (PEFT).** The goal of PEFT methods is to reduce the computational resources (memory footprint, wall time, etc.) required for fine-tuning large models such as Transformers (Vaswani et al., 2017) and Vision Transformers (ViTs) (Dosovitskiy et al., 2021). Prompt-based methods either append prompts to the input tokens (Lester et al., 2021; Jia et al., 2022), or the intermediate layers (Li & Liang, 2021; Liu et al., 2021; Zhang et al., 2023c). Early work on PEFT used sequential/parallel learnable adapters added after the Multi-Head Self Attention and/or FFN blocks (Houlsby et al., 2019; Bapna & Firat, 2019; Pfeiffer et al., 2021; 2020; Rücklé et al., 2021; Mahabadi et al., 2021a; Chen et al., 2022). LoRA (Hu et al., 2022) and its variants (Zhang et al., 2023b; Dettmers et al., 2023; Lialin et al., 2023; Jie & Deng, 2023; Kopiczko et al., 2023; Gao et al., 2024; Liu et al., 2024) learn residuals to the pretrained weight matrices in the form of low-rank factorization, removing the added inference cost in adapter based methods. Tied LoRA (Renduchintala et al., 2024) shares the residual weights across layers, and also across Query, Key and Value components. BitFit (Zaken et al., 2022) fine-tunes all the bias terms in a pretrained backbone. MEND (Mitchell et al., 2022) edits a pretrained model by learning fine-tuning weights from the gradient inputs with a low-rank MLP parameterization.

**Hypernetworks.** Ha et al. (2016) introduced Hypernetworks, i.e., neural networks that generate the parameters for other neural networks, in language modeling tasks by generating the weights of an LSTM (Hochreiter & Schmidhuber, 1997). Hypernetworks have previously been applied for few-shot classification (Zhao et al., 2020; Zhmoginov et al., 2022), transfer learning (Requeima et al., 2019) and continual learning (von Oswald et al., 2020; Yin et al., 2022). Similar to our proposed approach, (Requeima et al., 2019) learns to adapt a global feature extractor through an adaptation network. In a few shot continual learning setup, (Vladymyrov et al., 2023) uses a hyper-Transformer to generate the parameters for a separate Convolutional Neural Network (ConvNet), which use as inputs both a support set of images of the current task and the ConvNet parameters generated for the previous tasks. HyperFormer++ (Mahabadi et al., 2021b) uses a Multi-Layer Perceptron (MLP) to generate the parameters from layer embedding and a latent vector for Adapters (Houlsby et al., 2019) introduced across layers of a pretrained model in a multitask setting. Unlike (Mahabadi et al., 2021b), we directly use the weights of the frozen pretrained model, thus eliminating the need for embeddings.

**Neural Functionals**: Our approach is related to neural functionals that aim to learn deep neural networks acting on the weights of other neural networks. For toy problems, equivariant architectures have been explored for tasks like classifying implicit neural representations (Navon et al., 2023; Zhou et al., 2023a;b; Kofinas et al., 2024), adapting model architectures to new domains (Navon et al., 2023), predicting model generalization performance (Zhou et al., 2023a;b; Kofinas et al., 2024; Lim et al., 2023), and learned optimizers (Zhou et al., 2024). However, our work is the first to explore fine-tuning of a model using it's own weights. We do not use equivariant architectures, but note that this direction of work is orthogonal to ours, and can be further explored in the future.

## 6 CONCLUSION

We present Generative Parameter Efficient Fine-Tuning (GIFT) for adapting pretrained Transformer backbones on downstream tasks. Our GIFT learns to generate the fine-tuning weight-residuals for layers selected in fine-tuning directly from their frozen pretrained weights using a neural network. We show that a simple design - where GIFT consists of two linear layers without bias terms - can achieve strong performance, which provides a novel angle for formulating PEFT methods. We further show that the simple GIFT bridges PEFT and ReFT methods. We conduct experiments across various tasks, including Natural Language Generation (instruction following, commonsense reasoning, and arithmetic reasoning), Natural Language Understanding, and Visual Recognition. GIFT outperforms previous PEFT methods while using approximately 14 times fewer parameters and surpasses previous ReFT approaches with half the parameters.

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

# APPENDIX

## A  ADDITIONAL EXPERIMENTS

### A.1  LANGUAGE UNDERSTANDING ON GLUE

Table 7: Results on the GLUE benchmark. Following the common protocol, we report the Matthew's Correlation for CoLA, Pearson's Correlation for STS-B. For all other datasets, we report the accuracy. The preceding superscript, if added, indicates the source of results.

| Method | Params (%) | SST-2 | MRPC | CoLA | QNLI | RTE | STS-B | Avg. |
|---|---|---|---|---|---|---|---|---|
| **RoBERTa-Base** | | | | | | | | |
| [Hu et al., 2022]FT | 100 | 94.8 | 90.2 | 63.6 | 92.8 | 78.7 | 91.2 | 85.2 |
| [Hu et al., 2022]BitFit | 0.080 | 93.7 | 92.7 | 62.0 | 91.8 | 81.5 | 90.8 | 85.4 |
| [Hu et al., 2022]Adpt$^D$ | 0.240 | $94.2_{\pm0.1}$ | $88.5_{\pm1.1}$ | $60.8_{\pm0.4}$ | $93.1_{\pm0.1}$ | $71.5_{\pm2.7}$ | $89.7_{\pm0.3}$ | 83.0 |
| [Hu et al., 2022]Adpt$^D$ | 0.720 | $94.7_{\pm0.3}$ | $88.4_{\pm0.1}$ | $62.9_{\pm0.9}$ | $93.0_{\pm0.2}$ | $75.9_{\pm2.2}$ | $90.3_{\pm0.1}$ | 84.2 |
| [Kopiczko et al., 2023]LoRA (Hu et al., 2022) | 0.240 | $95.1_{\pm0.2}$ | $89.7_{\pm0.7}$ | $63.4_{\pm1.2}$ | $93.3_{\pm0.3}$ | $86.6_{\pm0.7}$ | $91.5_{\pm0.2}$ | **86.6** |
| VeRA (Kopiczko et al., 2023) | **0.034** | $94.6_{\pm0.1}$ | $89.5_{\pm0.5}$ | $65.6_{\pm0.8}$ | $91.8_{\pm0.2}$ | $78.7_{\pm0.7}$ | $90.7_{\pm0.2}$ | 85.2 |
| GIFT$_{Q,V}^{32}$ | 0.079 | $94.8_{\pm0.3}$ | $90.0_{\pm1.1}$ | $64.1_{\pm1.0}$ | $92.7_{\pm0.2}$ | $78.7_{\pm2.1}$ | $90.3_{\pm0.1}$ | 85.1 |
| **RoBERTa-Large** | | | | | | | | |
| [Hu et al., 2022]Adpt$^P$ | 0.847 | $96.1_{\pm0.3}$ | $90.2_{\pm0.7}$ | $68.3_{\pm1.0}$ | $94.8_{\pm0.2}$ | $83.8_{\pm2.9}$ | $92.1_{\pm0.7}$ | 87.6 |
| [Hu et al., 2022]Adpt$^P$ | 0.226 | $96.6_{\pm0.2}$ | $89.7_{\pm1.2}$ | $67.8_{\pm2.5}$ | $94.8_{\pm0.3}$ | $80.1_{\pm2.9}$ | $91.9_{\pm0.4}$ | 86.8 |
| [Hu et al., 2022]Adpt$^H$ | 1.693 | $96.2_{\pm0.3}$ | $88.7_{\pm2.9}$ | $66.5_{\pm4.4}$ | $94.7_{\pm0.2}$ | $83.4_{\pm1.1}$ | $91.0_{\pm1.7}$ | 86.8 |
| [Hu et al., 2022]Adpt$^H$ | 0.226 | $96.3_{\pm0.5}$ | $87.7_{\pm1.7}$ | $66.3_{\pm2.0}$ | $94.7_{\pm0.2}$ | $72.9_{\pm2.9}$ | $91.5_{\pm0.5}$ | 84.9 |
| [Zhang et al., 2023a]LoRA-FA | 1.044 | 96.0 | 90.0 | 68.0 | 94.8 | 86.1 | 92.0 | 87.7 |
| [Kopiczko et al., 2023]LoRA (Hu et al., 2022) | 0.226 | $96.2_{\pm0.5}$ | $90.2_{\pm1.0}$ | $68.2_{\pm1.9}$ | $94.8_{\pm0.3}$ | $85.2_{\pm1.1}$ | $92.3_{\pm0.5}$ | **87.8** |
| VeRA (Kopiczko et al., 2023) | **0.017** | $96.1_{\pm0.1}$ | $90.9_{\pm0.7}$ | $68.0_{\pm0.8}$ | $94.4_{\pm0.2}$ | $85.9_{\pm0.7}$ | $91.7_{\pm0.8}$ | **87.8** |
| GIFT$_{Q,V}^{32}$ | 0.037 | $95.8_{\pm1.1}$ | $88.7_{\pm1.2}$ | $67.0_{\pm1.5}$ | $94.7_{\pm0.1}$ | $87.0_{\pm1.4}$ | $91.5_{\pm0.8}$ | 87.5 |

**Data.** General Language Understanding Evaluation benchmark (GLUE) (Wang et al., 2018) is a widely used benchmark for sequence classification, where the model must learn to classify the entire sentence into two categories. We finetune RoBERTa-Base/Large models (Liu et al., 2019) with the pretrained checkpoints from HuggingFace using GIFT. We also compare with VeRA (Kopiczko et al., 2023). We follow a similar experimental setup as VeRA (Kopiczko et al., 2023): We do not evaluate on MNLI and QQP tasks due to computational budget restrictions, and hence do not use the MNLI trick as done in LoRA (Hu et al., 2022) [2]. Our hyperparameters are provided in the Appendix C.6.

**Results.** Table 7 shows the results. our GIFT achieves similar performance as prior PEFT methods. We note that although VeRA obtains slightly better performance than our GIFT using less parameters, the randomly initialized and frozen $\mathbf{A}$ and $\mathbf{B}$, VeRA does not scale to larger models and more challenging tasks, as seen in Table 2 and Table 3 in the main text. We hypothesize that when a downstream task is out of distribution to the pretraining those randomly initialized $A$ and $B$ may have limited expressivity.

### A.2  VTAB-1K

Table 8: Results on the VTAB-1k benchmark (Zhai et al., 2019). #Params and GPU Memory are reported in the same way as those in Table 4.

| Method | Params (%) ↓ | GPU Mem (G) ↓ | Natural | Specialized | Structured | *Avg* |
|---|---|---|---|---|---|---|
| VPT (Jia et al., 2022) | 0.054 | 2.753 | 81.03 | 85.65 | 58.89 | 72.68 |
| BitFit (Zaken et al., 2022) | 0.097 | 2.673 | 81.79 | 85.15 | 57.75 | 72.37 |
| LoRA (Hu et al., 2022) | 0.172 | **2.622** | **81.96** | 85.89 | 60.98 | 73.95 |
| GIFT$_Q^{16}$ | **0.029** | 2.644 | 81.95 | **86.30** | **61.12** | **74.09** |

### A.3  CHOICE OF FINE-TUNING THE PROJECTION LAYER IN MHSA FOR VISION TASKS.

As opposed to the common convention of applying PEFT methods to the Query and Value layers, we apply GIFT to the last linear projection layer of the Multi-Head Self Attention block, following (Savadikar et al., 2023) who show that the projection layer is a sweet spot to for finetuning and growing model for continual learning. We empirically observe that applying GIFT to the Projection layer has two advantages: (1) it results in better visual clusters (2) achieves almost the same performance as

---

[2]For experiments with RoBERTa-Base, LoRA (Hu et al., 2022) use the checkpoints from the MNLI task when fine-tuning on the RTE, STS-B and MRPC tasks.

that of the Query and Value layers at half the parameter cost, as seen in Table 9. We also note that in both cases, GIFT performs better than LoRA.

Table 9: Comparisons between the projection layer in the MHSA module and the Query+Value layer in PEFT using LoRA and our GIFT on the FGVC benchmark.

| Component | Method | #Params (M) ↓ | GPU Mem (G) ↓ | CUBS | Bird | Flower | Dog | Car | *Avg* |
|---|---|---|---|---|---|---|---|---|---|
| Proj | LoRA | 0.147 | 2.62 | 88.00 | 84.94 | **99.32** | 85.36 | **85.92** | 88.71 |
|  | GIFT | 0.025 | 2.65 | **89.71** | **86.28** | 99.22 | **87.44** | 84.28 | **89.39** |
| Q+V | LoRA | 0.295 | 2.97 | 87.97 | 84.85 | 99.20 | 84.62 | **87.03** | 88.73 |
|  | GIFT | 0.049 | 2.97 | **89.54** | **86.47** | **99.45** | **86.92** | 85.42 | **89.56** |

## B  GRADIENT ANALYSIS

During fine-tuning, the derivatives to the learnable parameters $B^l$ and $A^l$ in LoRA and those to $\phi$ and $\psi$ in GIFT are significantly different. Without loss of generality, consider a toy isotropic MLP with 2 hidden layers and no bias terms as the pretrained backbone:

$$\textbf{Toy MLP:} \quad x^1 = \sigma\left(x^0 W^{1\top}\right), x^2 = \sigma\left(x^1 W^{2\top}\right), x^3 = x^2 W^{3\top}, \tag{11}$$

where $x^0, x^1, x^2, x^3 \in \mathbb{R}^{N \times d}$ are the hidden representations of an $N$-token sequence in the $d$-dim space, $W^1, W^2, W^3 \in \mathbb{R}^{d_{out} \times d_{in}}$ denote the pretrained weights, and $\sigma(\cdot)$ denotes the activation function. For simplicity, we assume that $d_{out} = d_{in} = d$.

Consider fine-tuning layers 1 and 3. Let $\ell$ be the scalar loss (e.g., the cross-entropy loss) computed using one data point for simplicity. For LoRA at the first layer, we have,

$$\textbf{LoRA:} \quad \frac{\partial \ell}{\partial A^1} = B^{1\top} \cdot x'_1, \qquad \frac{\partial \ell}{\partial B^1} = x'_1 \cdot A^{1\top}; \quad \text{where } x'_1 = \frac{\partial \ell}{\partial x^1}^\top \cdot x^0, x'_1 \in \mathbb{R}^{d \times d}, \tag{12}$$

and for $\psi$ and $\phi$ shared by layers 1 and 3 in our GIFT,

$$\textbf{GIFT:} \quad \frac{\partial \ell}{\partial \psi} = \phi^\top \cdot y'_{1,3}, \qquad \frac{\partial \ell}{\partial \phi} = y'_{1,3} \cdot \psi^\top; \quad \text{where } y'_{1,3} = W^{1\top} \cdot x'_1 + W^{3\top} \cdot x'_3, \tag{13}$$

where $\psi$ and $\phi$ gather information from all the selected layers in a way more holistic than LoRA, e.g., comparing $y'_{1,3}$ vs $x'_1$.

## C  IMPLEMENTATION DETAILS AND HYPERPARAMETER TUNING

In practice, we use a scaling factor of $\frac{\alpha}{r}$ for residuals as done in LoRA (Hu et al., 2022):

$$\hat{W}^l_{d_{out} \times d_{in}} = W^l_{d_{out} \times d_{in}} + \frac{\alpha}{r} W^l_{d_{out} \times d_{in}} \cdot \phi_{d_{in} \times r} \cdot \mathbb{A}_{r \times d_{in}}, \tag{14}$$

We omit this in the main section for ease of notation and simplicity, as it does not affect the analysis. In experiments, we initialize $\psi$ to all zeros and $\phi$ to Kaiming Uniform initialization (He et al., 2015).

### C.1  COMPUTING RESOURCES AND CODE

All our experiments are run on a single Nvidia A100 GPU. Our code is provided in the supplementary materials.

### C.2  INSTRUCTION FOLLOWING

Following Wu et al. (2024b), we finetune LLaMA-1 7B (Touvron et al., 2023a) using the Alpaca52k dataset (Taori et al., 2023) and evaluate using GPT4 Turbo as the annotator during the hyperparameter search. After finding the best hyperparameters, we finetune Llama-2 7B (Touvron et al., 2023b) using the Ultrafeedback dataset and use GPT4 as the annotator with Alpaca-Eval 1.0. This setup prevents overfitting to the hyperparameters and GPT4 as the judge. Table 10 shows the hyperparameters used in our experiments. We report the final results by finetuning the Llama-2 model Touvron et al. (2023b) using the Ultrafeedback dataset Cui et al. (2023) and use and use Alpaca-Eval v1.0 Li et al. (2023) with GPT-4 as the annotator for evaluation. We report average results of 2 runs with seeds 42 and 43.

We use the same inference strategy as Wu et al. (2024b) and Wu et al. (2024a) to ensure fair comparison: we use a greedy decoding strategy, maximum repetition penalty of 1.1, maximum repetition n-gram size of 5, and maximum new token number of 2048.

Table 10: Hyperparameters used for the Instruction Tuning experiments. We only perform search over the learning rate and the rank, and choose the other hyperparameters from Wu et al. (2024b). Bold faced and underlined pairs denote the final hyperparameters.

| Hyperparameter | Value |
|---|---|
| Learning Rate | $5e^{-5}, 7.5e^{-5}, 1e^{-4}, 2.5e^{-4}, \underline{5e^{-4}}, 7.5e^{-4}, \mathbf{1e^{-3}}$ |
| Rank | $\mathbf{16}, 32, 64, \underline{128}$ |
| Optimizer | AdamW (Loshchilov & Hutter, 2019) |
| Weight Decay | 0.0 |
| LR Scheduler | Linear |
| Warmup Ratio | 0.0 |
| Batch Size | 4 |
| Gradient Accumulation Steps | 32 |
| Epochs | 12 |

## C.3 ARITHMETIC REASONING

Following Wu et al. (2024b), we tune the hyperparameters by fine-tuning the LLaMA-1 (7B) model on the GSM8k dataset Cobbe et al. (2021) using a separate validation set constructed from the training set, and use the same hyperparamters for LLaMA-1 (13B), Llama-2 (7B) and Llama-3 (8B). Table 11 shows the hyperparameters used in our experiments. We perform hyperparameter search using the seed 42, and report the final results by averaging over three runs with seeds 42, 43, and 44. We use a greedy decoding scheme during inference, with a maximum new token number of 512.

Table 11: Hyperparameters used for the arithmetic reasoning experiments. The final hyperparameters are underlined.

| | Hyperparameter | Value |
|---|---|---|
| | Max Sequence Length | 512 |
| | Optimizer | AdamW |
| | Weight Decay | 0.0 |
| | LR Scheduler | Linear |
| | Batch Size | 16 |
| | Gradient Accumulation Steps | 1 |
| | Epochs | 3 |
| $\text{GIFT}^{64}_{\underline{Q},\underline{K},\underline{V},\underline{U},\underline{D}}$ | Learning Rate | $\{1, 2, 3, 4, 5, \underline{6}, 7, 8, 9, 10\} \times 10^{-4}$ |
| | Rank | 64 |
| | Scaling Factor | $\underline{64}, 128$ |
| | Warmup Ratio | $\underline{0.06}, 0.1$ |
| $\text{GIFT}^{64}_{\overline{O},\overline{D}}$ | Learning Rate | $\{1, 2, 3, 4, 5, 6, 7, \underline{8}, 9, 10\} \times 10^{-4}$ |
| | Rank | 64 |
| | Scaling Factor | $\underline{64}, 128$ |
| | Warmup Ratio | $0.06, \underline{0.1}$ |
| $\text{GIFT}^{16}_{\underline{QKV},\overline{O},\underline{UG},\overline{D}}$ | Learning Rate | $\{1, 2, 3, 4, \underline{5}, 6, 7, 8, 9, 10\} \times 10^{-4}$ |
| | Rank | 16 |
| | Scaling Factor | $16, \underline{32}$ |
| | Warmup Ratio | $\underline{0.06}, 0.1$ |
| VeRA | Learning Rate | $\{2, 3, 4, 5, 6, 7, 8, 9, 10, \underline{20}, 30\} \times 10^{-3}$ |
| | Rank | 12288 |
| | Warmup Ratio | $\underline{0.06}, 0.1$ |

## C.4 COMMONSENSE REASONING

We tune the hyperparameters for commonsense reasoning by fine-tuning the LLaMA-1 model on the BoolQ dataset Clark et al. (2019) using a separate validation set constructed from the training set. Table 12 shows the hyperparameters used in our experiments. We search for the hyperparameters using LLaMa-1 (7B) and use the same hyperparameters for LLaMA-1 (13B), Llama 2 (7B) and Llama 3 (8B) models. We perform hyperparameter search using the seed 42, and report the final

results by averaging over three runs with seeds 42, 43, and 44. We use a greedy decoding scheme during inference, with a maximum new token number of 32.

Table 12: Hyperparameters used for the commonsense reasoning experiments. The final hyperparameters are underlined.

|  | Hyperparameter | Value |
|---|---|---|
|  | Max Sequence Length | 512 |
|  | Optimizer | AdamW |
|  | Weight Decay | 0.0 |
|  | LR Scheduler | Linear |
|  | Batch Size | 16 |
|  | Gradient Accumulation Steps | 1 |
|  | Epochs | 3 |
| $\text{GIFT}^{64}_{\underline{Q},\underline{K},\underline{V},\underline{U},\underline{D}}$ | Learning Rate | $\{0.7, 0.8, 0.9, \underline{1}, 2, 3, 4, 5, 6, 7\} \times 10^{-4}$ |
|  | Rank | 64 |
|  | Scaling Factor | 64, $\underline{128}$ |
|  | Warmup Ratio | 0.06, $\underline{0.1}$ |
| $\text{GIFT}^{64}_{\overline{O},\overline{D}}$ | Learning Rate | $\{0.7, 0.8, 0.9, 1, 2, 3, 4, 5, \underline{6}, 7\} \times 10^{-4}$ |
|  | Rank | 64 |
|  | Scaling Factor | 64, $\underline{128}$ |
|  | Warmup Ratio | $\underline{0.06}$, 0.1 |
| $\text{GIFT}^{16}_{\underline{QKV},\overline{O},\underline{UG},\overline{D}}$ | Learning Rate | $\{0.7, 0.8, 0.9, \underline{1}, 2, 3, 4, 5, 6, 7\} \times 10^{-4}$ |
|  | Rank | 16 |
|  | Scaling Factor | 16, $\underline{32}$ |
|  | Warmup Ratio | $\underline{0.06}$, 0.1 |
| VeRA | Learning Rate | $\{5, 6, 7, \underline{8}, 9, 10, 20\} \times 10^{-3}$ |
|  | Rank | 4096 |
|  | Warmup Ratio | 0.06, $\underline{0.1}$ |

## C.5 FGVC AND VTAB EXPERIMENTS

For all the experiments, we use ViT-B/16 model Dosovitskiy et al. (2021), which contains 12 transformer blocks, each with 12 heads in the Multi-Head Self-Attention (MHSA) blocks, and a dimension of 768. We use checkpoints from the model pretrained on the ImageNet21k Deng et al. (2009) under the supervised training protocol provided by the `timm` package. For both VTAB and FGVC experiments, we use a hyperparameter search using the validation sets and use the training+validation data during the final run and report the results on the test sets. The hyperparameter search space used in our experiments in provided in Table 13. For the VTAB benchmark, we use the official splits provided by Zhai et al. (2019). For the FGVC benchmark, we use the same train, validation and test splits as Shi et al. (2023), *except for* Stanford Cars dataset Gebru et al. (2017). Due to the unavailability of the dataset from the original source, and the difference in the format of the data provided by the updated source, we create our own training and validation split (with the same number of images as Shi et al. (2023)) and use the official testing split. We initialize $\phi$ with zeros and $\psi$ with Kaiming uniform initialization.

## C.6 GLUE BENCHMARK

For our experiments with the GLUE benchmarks, we use RoBERTa-Base and RoBERTA-Large models Liu et al. (2019). Table 14 shows the hyperparameters used in our experiments. We perform the hyperparameter search using a single seed, and use 5 seeds for the final run and report the median across them. We use the seed 42 for performing hyperparameter search and use 42, 43, 44, 45, 46 for the final runs. We observe that for RoBERTa-Large, the training is unstable for some seeds, and hence replace them. Following Hu et al. (2022) and Kopiczko et al. (2023), we use the training split for training the models and report the results on the validation split. We initialize $\phi$ with zeros and $\psi$ with Kaiming uniform initialization.

Table 13: Hyperparameter search space used for GLUE experiments. During the search, we use 50 epoch for each experiment in the VTAB benchmark for 25 epochs for the FGVC benchmark due to computational constraints, and use 100 epochs in the final run with the selected hyperparameters

|  | Hyperparameter | Values |
|---|---|---|
| BitFit | Learning Rate | $1e^{-3}, 1.5e^{-3}, 2e^{-3}, 2.5e^{-3}, 5e^{-3}, 1e^{-2}$ |
|  | Weight Decay | 0.0 |
| VPT | Learning Rate | $1e^{-3}, 1.5e^{-3}, 2e^{-3}, 2.5e^{-3}, 5e^{-3}, 1e^{-2}$ |
|  | Weight Decay | 0.0 |
|  | Num. Prompts | 5 |
| LoRA | Learning Rate | $1e^{-3}, 1.5e^{-3}, 2e^{-3}, 2.5e^{-3}, 5e^{-3}, 1e^{-2}$ |
|  | Weight Decay | 0.01, 0.001, 0.0001, 0.0 |
|  | Rank $r$ | 8 |
| GIFT | Learning Rate | $1e^{-4}, 2.5e^{-4}, 5e^{-4}, 1e^{-3}, 2.5e^{-3}, 5e^{-3}$ |
|  | Weight Decay | 0.01, 0.001, 0.0001, 0.0 |
|  | Rank $r$ | 16 |
|  | Optimizer | AdamW |
|  | LR Scheduler | Cosine |
|  | Warmup Epochs | 5 |
|  | Epochs | 100 |
|  | Batch Size | 32 |

Table 14: Hyperparameter search space used for GLUE experiments. Except for $r$, learning rate and weight decay, all the other hyperparameters have been taken from Kopiczko et al. (2023) due to computational constraints.

|  | Hyperparameters | SST-2 | MRPC | CoLA | QNLI | RTE | STS-B |
|---|---|---|---|---|---|---|---|
|  | Optimizer | | | AdamW | | | |
|  | Warmup Ratio | | | 0.06 | | | |
|  | LR Scheduler | | | Linear | | | |
|  | Rank $r$ | | | 32 | | | |
| Base | LR | $5e^{-4}$ | $1e^{-3}$ | $5e^{-4}$ | $5e^{-4}$ | $7.5e^{-4}$ | $5e^{-4}$ |
|  | Weight Decay | 0 | 0 | $1e^{-4}$ | $1e^{-2}$ | $1e^{-3}$ | $1e^{-3}$ |
|  | Epochs | 60 | 30 | 80 | 25 | 160 | 80 |
|  | Batch Size | | | 64 | | | |
|  | Max Seq. Len. | | | 512 | | | |
| Large | LR | $1e^{-3}$ | $5e^{-4}$ | $7.5e^{-4}$ | $2.5e^{-4}$ | $5e^{-4}$ | $7.5e^{-4}$ |
|  | Weight Decay | $1e^{-2}$ | $1e^{-4}$ | $1e^{-2}$ | 0 | $1e^{-2}$ | 0 |
|  | Epochs | 10 | 40 | 40 | 20 | 40 | 20 |
|  | Batch Size | | | 128 | | | |
|  | Max Seq. Len. | | | 128 | | | |

## D  GENERATION EXAMPLES

Commonsense Reasoning (WinoGrande)

Please choose the correct answer to fill in the blank to complete the given sentence: Sarah was a much better surgeon than Maria so _ always got the easier cases.

Option1: Sarah Option2: Maria Answer format: option1/option2

- - - - - - - - - - - - - - - - - - - - - - - - - - - - - - - - - - - - - - - - - - -

| Model | Response |
|---|---|
| **LLaMA-1 7B (GIFT)** | option1 |
| **Llama-2 7B (GIFT)** | option2 |
| **Llama-3 7B (GIFT)** | option2 |

**Commonsense Reasoning (OBQA)**

Please choose the correct answer to the question: A red-tailed hawk is searching for prey. It is most likely to swoop down on

Answer1: an eagle Answer2: a cow Answer3: a gecko Answer4: a deer

Answer format: answer1/answer2/answer3/answer4

- - - - - - - - - - - - - - - - - - - - - - - - - - - - - - - - - - - - - - - - - - - - - - - - - - -

| Model | Response |
|---|---|
| **LLaMA-1 7B (GIFT)** | answer3 |
| **Llama-2 7B (GIFT)** | answer4 |
| **Llama-3 7B (GIFT)** | answer3 |

**Arithmetic Reasoning (GSM8K)**

A car in the fast lane is traveling at 60 miles/hour. A car in the slow lane is traveling at half that speed. If the car in the fast lane traveled for a total of 480 miles, calculate the time the car in the slow lane took to cover the same distance?

| Model | Response |
|---|---|
| **LLaMA-1 7B** | The car in the slow lane is traveling at 60/2 = 30 miles/hour |

Let's assume the time taken by the car in the fast lane to cover 480 miles is t.
Distance = Speed x Time
Distance in fast lane = 60t
Distance in slow lane = 30t
We know that the distance covered by the car in the fast lane is 480 miles.
Therefore, 60t = 480
t = 480/60
t = 8 hours
Now, we need to calculate the time taken by the car in the slow lane to cover the same distance.
Distance in slow lane = 30t
480 = 30t
t = 480/30
t = 16 hours
Therefore, the car in the slow lane took 16 hours to cover the same distance as the car in the fast lane.
Answer: 16.

**Instruction Following (Alpaca-Eval) - Llama-2 7B**

List the layers of the TCP/IP model and for each layer identify one protocol.

| Model | Response |
| --- | --- |
| **GIFT (r=128)** | The TCP/IP (Transmission Control Protocol/Internet Protocol) model is a standard network architecture used to communicate between devices on the Internet. It consists of four main layers, which are as follows:
1. Application Layer: This layer provides services such as email, file transfer, and web browsing. The most common protocol at this layer is HTTP (Hypertext Transfer Protocol).
2. Transport Layer: This level handles data transmission between applications. The most common transport protocols include TCP (Transmission ControlProtocol) and UDP (User Datagram Protocol).
3. Internet Layer: This is responsible for routing packets across the internet. The most common internet protocol at this layeris IP (Internet Protocol).
4. Network Interface Layer: This interface connects the computer to the physical network. The most common network interface protocol is Ethernet. |
| **GIFT (r=16)** | The TCP/IP (Transmission Control Protocol / Internet Protocol) model is a standard network architecture used to communicate between devices on the internet or any other computer network. It consists of four layers, each with its own set of protocols:
1. **Application Layer** - This layer provides services such as email, web browsing, file transfer, and remote login. The most common protocol at this layer is Hypertext Transfer Protocol (HTTP).
2. **Transport Layer** - It ensures reliable data delivery by segmenting messages into smaller units called segments and reassembling them at the receiving end. The main protocol here is Transmission Control Protocol (TCP).
3. **Internet Layer** - Also known as the Network Layer, it handles routing and forwarding of packets across multiple networks. The primary protocol in this layer is Internet Protocol (IP).
4. **Data Link Layer** - Responsible for establishing and maintaining connections between two hosts on a network. The most commonly used protocol at this level is Ethernet.
Here are some examples of protocols associated with each layer:
- Application Layer: HTTP, FTP, Telnet, SMTP, DNS, etc.
- Transport Layer: TCP, UDP
- Internet Layer: IPv4, IPv6
- Data Link Layer: Ethernet, Token Ring, FDDI, ATM, etc. |

---

**Instruction Following (Alpaca-Eval) - Llama-2 7B**

Write a pitch for a horror movie that plays on the island Sylt in northern germany in the 19th century. Include fishermen and a sinister force that drives the inhabitants insane.

- - - - - - - - - - - - - - - - - - - - - - - - - - - - - - - - - - - - - - - - - - - - - - -

| **Model** | **Response** |
|---|---|
| **GIFT (r=128)** | Title: "The Curse of Sylt" 
 Synopsis: A group of fishermen from the picturesque island of Sylt, located in Northern Germany, find themselves caught in a web of terror as they uncover a sinister force lurking beneath the surface of their tranquil community. As the men delve deeper into the mystery, they begin to experience strange visions and hear whispers in the night, leading them down a path of madness and despair. The island's inhabitants are driven to the brink of insanity by this malevolent force, which seems to have no end in sight. Can our heroes break free from its grasp before it's too late? Find out in "The Curse Of Sylt," a chilling tale of horror and suspense set against the backdrop of a beautiful but deadly island paradise. |
| **GIFT (r=16)** | Title: "The Haunting of Sylt" 
 Synopsis: A group of fishermen from the small village of List on the remote island of Sylt, Germany, are plagued by a mysterious force that drives them to commit unspeakable acts of violence against their loved ones. As the villagers succumb to madness, they begin to suspect that something sinister lurks beneath the surface of the island's tranquil waters. Is it a curse? Or could there be a more sinister explanation for the chaos that has descended upon their once-peaceful community? Join us as we delve into the depths of this haunted island and discover the truth behind its dark secrets. |

# E    VISUAL INSPECTION OF OUR TWO-LINEAR-LAYER PARAMETERIZED GIFT

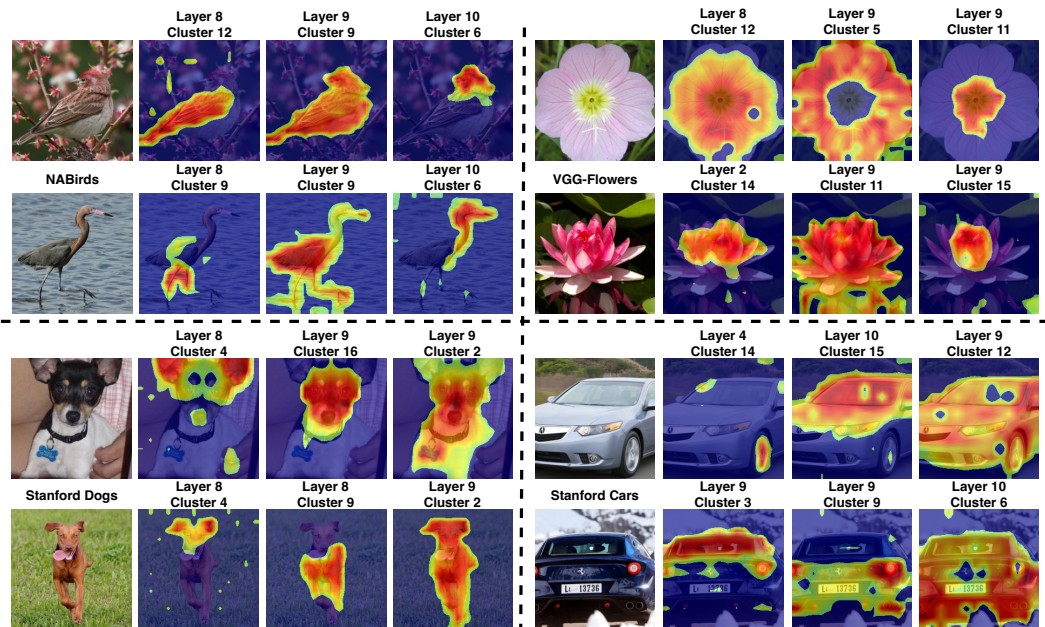

Figure 5: More examples of the visual interpretability of our two-linear-layer parameterized GIFT tested on the FGVC benchmark. We show examples of head, wings and legs of birds in the *top-left*, examples of flower petals in the *top-right*, examples of head, ears and legs of dogs in the *bottom-left*, and examples of tires, windshield and bumper of cars in the *bottom-right*.

