# OpenReview forum: "Generative Parameter Efficient Fine-Tuning"
_ICLR.cc/2025/Conference — Submitted to ICLR 2025_

### Official Review · Reviewer_53Qz · 2024-10-30

**Soundness:** 3
**Presentation:** 1
**Contribution:** 3
**Rating:** 5
**Confidence:** 4

**Summary:**

This paper proposes a new fine-tuning method called Generative Parameter-Efficient Fine-tuning (GIFT), which trains two linear layers to project the pre-trained weight matrices into fine-tuned weights. The authors argue that it offers a unifying perspective on PEFT and representation-efficient fine-tuning (ReFT) approaches by projecting pre-trained weights linearly. The results demonstrate that GIFT improves performance while using fewer parameters compared to previous parameter-efficient fine-tuning (PEFT) methods, such as LoRA, ReFT, and VeRA.

**Strengths:**

1. This paper provides new insights into fine-tuning techniques, suggesting that the proposed Generative Parameter-Efficient Fine-Tuning (GIFT). I think it can be viewed as a specific form of Representation Fine-Tuning (ReFT), where all tokens in the selected layers share the same re-parameterization parameters. Notably, GIFT is easier to implement than ReFT, utilizing two linear layers for weight re-parameterization without explicitly modifying token embeddings.

2. The performance is impressive. They achieve similar or better performance with fewer parameters comparing to the other PEFT. Validation was conducted across multiple tasks, datasets, and models, demonstrating GIFT's effectiveness and versatility.

**Weaknesses:**

1. In my opinion, the authors make claims about aspects that remain unexplained, which can confuse readers. The authors should provide further clarification to support these claims. For instance:

(Line 051) "but the learnable weight-residuals do not have direct information exchange with the pre-trained weights"

(Line 135) "one of the simplest updates that minimally distorts and maximally preserves the pre-trained knowledge is defined by Eqn.1 and Eqn.2, thanks to the low-rank factorized linear projection in the parameter space."

(Line 181) "Additionally, adding fixed random matrices with learnable scales in PEFT makes the relationship between fine-tuned models and frozen pretrained models less intuitive."

(Line 194) "Furthermore, token-level interventions lack a holistic understanding of the relationship between ReFTed models and frozen pretrained models."

2. There are issues with mathematical notation throughout the paper, making it difficult for readers to follow the ideas presented. Please refer to other PEFT papers (e.g., ReFT, LoRA) for guidance on presenting mathematical concepts clearly. For example, matrices should be represented in bold uppercase, vectors in bold lowercase, and scalars in italic lowercase. Additionally, in machine learning literature, symbols like θ (theta) and φ (phi) are conventionally used to denote model parameters, not calculations within the model.

3. The content has redundancies; for instance, the Related Work section and Section 2.1 cover similar material, leading to overlap.

**Questions:**

See weaknesses.

+)

1. What did the authors want to say in Section 2.3? I think the idea behind this is similar to 2.2. What is the meaning of "accumulate" in the paragraph?

2. Please provide more details regarding the experiment in Fig. 2. Specifically, clarify the meaning of the cluster. If the authors aim to demonstrate that GIFT enhances object highlighting within the attention modules, results should be compared with those of the pretrained model for a meaningful evaluation.

---

> ### Author Response · Authors · 2024-11-21
> **Author Response**
>
> Thank your for your valuable feedback. Following are the clarifications on your comments:
>
> > Mathematical notation
>
> Thank you for your advice on improving the mathematical notation. We have revised the notation to make it more consistent and clear, which is reflected in equations (5) and (6) in the revised manuscript.
> - However, we retain the use of $\phi$ and $\psi$ to denote the learnable parameters, as they are a part of the parameters of the weight-generator network. This lets us distinguish between standard fine-tuning methods and GIFT. Our GIFT generates the fine-tuned weights from the pre-trained weights, rather than learning them as model parameters.
> - This also keeps the notation consistent across equations (5) and (6).
>
> > Content redundancy
>
> We have rephrased the Introduction (Section 1) and Approach (Section 2) to be more precise and concise. Please help skim those to see if we address your concerns.
>
> > Clarification on statements
>
> Thank you for your carefully reading through our submission. Those statements highlighted by you are indeed at a high level, indended to convey intuition behind GIFT. We have carefully rewritten some statements in the revised manuscript to be more precise and clear.
>
> > Line 051: but the learnable weight-residuals do not have direct information exchange with the pre-trained weights
>
> > Line 135: one of the simplest updates that minimally distorts and maximally preserves the pre-trained knowledge is defined by Eqn.1 and Eqn.2, thanks to the low-rank factorized linear projection in the parameter space.
>
> > Line 181: Additionally, adding fixed random matrices with learnable scales in PEFT makes the relationship between fine-tuned models and frozen pretrained models less intuitive
>
> We have replaced these statements to convey the message more clearly in Section 2.3 in the revised manuscript. We reiterate the modified statements from Section 2.3 here for convenience:
>
> "Pretrained Transformer backbones encode diverse knowledge from large-scale pretraining datasets within their weights. Fine-tuning them for a downstream task aims to incorporate new information from the task-specific training data and utilize the information present in the pretrained weights to the fullest extent. To achieve this, the fine-tuned weights can be directly conditioned on the pretrained weights, such that the new information is learned conditionally from the information in the pretrained weights. While LoRA and it's variants use a residual structure to address this, the residual weights are not directly conditioned on the pretrained weights, but rather learned via back-propagation (chain rule) updates. One of the simplest functions that can achieve this explicit conditioning is a linear transformation of the pretrained weights, as leveraged in Eqn. 7. Hence, the fine-tuned weights can also be expressed in the space of the pretrained weights $W_{d_{out}\times d_{in}}$ via $W_{d_{out}\times d_{in}}\cdot \Theta_{d_{in}\times d_{in}}$."
>
> > Line 194: "Furthermore, token-level interventions lack a holistic understanding of the relationship between ReFTed models and frozen pretrained models."
>
> The idea behind the statement is as follows: Although ReFT learns precise interventions for representations at token level, it is applied to fixed token position (prefix and suffix tokens), without regard to the actual token sequence. With GIFT's representation tuning perspective, the interventions can be more flexible since the tuning is applied to all the tokens. We have removed this statement, and instead rephrased the connection between GIFT and ReFT in Section 2.2 of the revised manuscript to convey the message more clearly.
>
> > Question 1
>
> Section 2.3 in the original manuscript describes the gradient of the learnable parameters in GIFT. The accumulation refers to the accumulation of the gradient across layers which occurs due to layerwise sharing of the learnable parameters. As this section is not critical to the understanding of the method, we have moved it to the appendix (Section B) in the revised version.
>
> > Question 2
>
> We have modified the description (Section 3.5) to be be more clear. We would like to point out that the segmentation maps are formed by projecting the output of the final projection layer in the MHSA block using the learned GIFT (equations 10 and 11 in the revised manuscript), and not the attention matrix.

---

> ### Author Response · Authors · 2024-11-22
> **Request your feedback and Appreciate your efforts**
>
> Dear Reviewer 53Qz,
>
> Hope all is well with you.
>
> We would like to request your feedback on our rebuttal, as well as the revised manuscript at your convenience.
>
> We look forward to it.
>
> Thank you very much.

---

> > ### Author Response · Authors · 2024-11-25
> > **Friendly reminder**
> >
> > Dear Reviewer 53Qz,
> >
> > Since the reviewer-author discussion will end tomorrow, we appreciate your initial comments, and would like request and look forward to your comment on our rebuttal and revised submission at your convince.
> >
> > Thank you very much.

---

> > > ### Comment · Reviewer_53Qz · 2024-11-27
> > >
> > > Thank you for your response.
> > >
> > > I believe the manuscript would be further enhanced if the authors referenced the notation used in this paper, which also employs a weight-generator network (https://arxiv.org/abs/2110.11309).

---

> ### Author Response · Authors · 2024-11-27
> **Thank you for your response**
>
> Dear Reviewer 53Qz,
>
> Thank you for pointing out this great work (MEND, Model Editor Networks with gradient Decomposition). We will discuss it in our revision.
>
> We would like to share our understanding with you in terms of the difference between the MEND method and our GIFT.
>
> >  **Summary of MEND and Its Notations**
>
> To edit layer $l$ with pretrained weights $W_l$, layer activation $u_l$ and output gradients $\delta_{l+1}$ are concatenate,  $z_l=\text{Concat}(u_l, \delta_{l+1})$, as input,  the editor network $g$ that  is parameterized as an MLP with low-rank weight matrices, $(U_1, V_1, U_2, V_2)$, residual connections and a single hidden layer with activation function $\sigma()$, and can be shared across layers with layer-specific scale $s_l$ and offset $o_l$,
>
> $g(z_l) = h_l + \sigma(s^2_l \odot U_2 V_2 h_l + o^2_l)$    (Eqn.3b in MEND)
>
> $h_l = z_l + \sigma(s^1_l \odot (U_1 V_1 z_l +b) + o^1_l)$   (Eqn.3a in MEND)
>
> The output of $g(z_l)$ is split into pseudoactivations $\tilde{u}_l$ and
>
> pseudodeltas $\tilde{\delta}_{l+1}$, the model edit for weight matrix $W_l$ is defined by,
>
>
> $\tilde{\nabla}_{W_l} =$
>
> $\sum_{i=1}^B {\tilde{\delta}^i_{l+1}} {\tilde{u}^i_l}^{\top}$      (We split the equations in two lines due to Markdown issues)
>
> The final edited weights are
>
> $\tilde{W}_l =$
>
> $W_l - \alpha \tilde{\nabla}_{W_l}$
>
> > **Notations in our GIFT (e.g. Eqns. 4 and 5)**
>
> $\hat{W}^l_{d_{out}\times d_{in}} = W^l_{d_{out}\times d_{in}} + \mathcal{G}(W^l_{d_{out}\times d_{in}}; \Theta)$
>
> $\qquad \qquad =W^l_{d_{out}\times d_{in}} +$ $W^l_{d_{out}\times d_{in}} \cdot \phi_{d_{in}\times r}\cdot \psi_{r\times d_{in}}$
>
> >> We use $W^l_{d_{out}\times d_{in}}$ to denote the pretrained weights, rather than $W_l$ used in MEND, since we want to explicitly show the dimensions of the weight matrix in the subscript.
>
> >> We use $\hat{W}$ to represent the final weights, rather than $\tilde{W}$ used in MEND.
>
> >>  We use $\phi_{d_{in}\times r}$ and $\psi_{r\times d_{in}}$ to denote the parameters of our GIFT weight-generator, rather than $U, V$ used in MEND.
>
> >> Overall, we think the notations used in our GIFT is self-contained and clear.  **We would like to hear your specific suggestions in terms of notation changes.**

---

> > ### Author Response · Authors · 2024-11-27
> > **Cite the suggested MEND work**
> >
> > Dear Reviewer 53Qz,
> >
> > We have cited MEND in Section 5 (line 453-454) in the revised manuscript.
> >
> > > MEND (Mitchell et al., 2022) edits a pretrained model by learning fine-tuning weights from the
> > gradient inputs with a low-rank MLP parameterization.
> >
> > Thanks.

---

### Official Review · Reviewer_k2rS · 2024-11-01

**Soundness:** 3
**Presentation:** 2
**Contribution:** 2
**Rating:** 5
**Confidence:** 3

**Summary:**

The paper introduces Generative Parameter-Efficient Fine-Tuning (GIFT), a method to fine-tune pretrained Transformer models with fewer parameters by generating fine-tuned weights from pretrained ones. They show this formulation can address the two questions about 1)an explicit and direct mapping between the fine-tuned model and the frozen pretrained model, 2) bridge  parameter-efficient fine-tuning and representation fine-tuning. The proposed GIFT method is designed by implementing a lightweight structure of only two linear layers, shared across selected layers in the model. Using minimal linear layers without bias, GIFT achieves significant parameter reductions compared to LoRA and performs better across some NLP and computer vision tasks, obtaining a slightly higher win rate on instruction tuning than GPT 3.5.

**Strengths:**

GIFT represents a unique approach to generating fine-tuned weights directly from pretrained weights, sharing parameters across layers to enhance efficiency.
Experiments demonstrate that GIFT outperforms existing PEFT methods on various natural language and computer vision tasks while using significantly fewer parameters, showing improvements in memory efficiency as well.
Tested on diverse tasks, GIFT shows effectiveness across commonsense reasoning, arithmetic, instruction following, and visual recognition tasks, reinforcing its versatility as a parameter-efficient fine-tuning approach.

**Weaknesses:**

Lack of some comparison settings on full finetuning:  How is the comparison results on full finetuning in Table-2 and Table-3 for commonsense reasoning and arithmetic reasoning task？Table-1 compares the full finetuning setting on Llama-2 7B for instruction following task, but table-2 and table-3 didn't reveal this setting.
Potential Scalability Concerns: Although parameter-efficient, the scalability of GIFT for larger models (beyond 8B parameters, like llama1-13B, llama3-65B) isn’t explicitly demonstrated. The experimental version of llama1-3 presented by the author is all less than or equal to 8B, leaving questions about its performance in high-scale deployment.
Limited Ablation on Layer Selection and Configuration: GIFT’s performance maybe vary depending on which layers are selected for fine-tuning. While some experiments address this, there is minimal ablation on different layer selection and the comparison between Lora with the same layers.
Some models compared are not newly advanced enough : Table-1 shows the result of fine-tuning Llama-2 7B with GIFT for instruction following task, but the compared gpt series is GPT 3.5 Turbo, a little bit outmoded. How is the comparison result of GIFT among some newly advanced models like gpt4o or others？

**Questions:**

See weaknesses.

---

> ### Author Response · Authors · 2024-11-21
> **Author Response 1/2**
>
> Thank you for your valuable feedback. Below, we provide clarifications on the points raised:
>
> > Lack of some comparison settings on full finetuning
>
> We acknowledge the value of such comparisons; however, due to resource limitations, we were unable to conduct experiments involving full fine-tuning of models. Please note that none of the prior works we compare against perform full fine-tuning. We have included the full fine-tuning baseline for instruction-following tasks where prior works have made these results available.
>
> > Potential Scalability Concerns
>
> - We have added experiments with LLaMA-1 13B for Commonsense Reasoning and Arithmetic Reasoning tasks. Table 2 and Table 3 in the revised manuscript show that GIFT performs better/on par with prior PEFT and ReFT methods while being significantly more parameter efficient. This shows that GIFT can be scaled to larger models on the scale of 13B, and can potentially be scaled to even larger models. We have summarized the results here for convenience.
>
> **Commonsense Reasoning (LLaMA-1 13B)**
> | **Method** | **Params (%)** | **BoolQ** | **PIQA** | **SIQA** | **HellaS.** | **WinoG.** | **ARC-e** | **ARC-c** | **OBQA** | **Avg**
> | - | - | - | - | - | - | - | - | - | - | -
> | PrefT | 0.031 | 65.3 | 75.4 | 72.1 | 55.2 | 68.6 | 79.5 | 62.9 | 68.0 | 68.4
> | AdapterS | 1.586 | 71.8 | 83.0 | 79.2 | 88.1 | 82.4 | 82.5 | 67.3 | 81.8 | 79.5
> | AdapterP | 2.894 | 72.5 | 84.9 | 79.8 | 92.1 | 84.7 | 84.2 | 71.2 | 82.4 | 81.5
> | LoRA | 0.670 | 72.1 | 83.5 | 80.5 | 90.5 | 83.7 | 82.8 | 68.3 | 82.4 | 80.5
> | DoRA | 0.681 | 72.4 | 84.9 | 81.5 | 92.4 | 84.2 | 84.2 | 69.6 | 82.8 | 81.5
> | DoRA (half) | 0.347 | 72.5 | 85.3 | 79.9 | 90.1 | 83.6 | 80.8 | 69.7 | 83.6 | 80.8
> | GIFT $^{64}_{\underline{Q},\underline{K},\underline{V},\underline{U},\underline{D}}$ | 0.034 | 74.3 | 87.3 | 81.8 | 95.3 | 86.5 | 87.4 | 76.2 | **89.0** | 84.7
> | DiReFT | 0.025 | 71.3 | 86.1 | 80.8 | 94.6 | 83.6 | 85.5 | 72.9 | 82.7 | 82.2
> | LoReFT | 0.025 | 72.1 | 86.3 | 81.8 | 95.1 | **87.2** | 86.2 | 73.7 | 84.2 | 83.3
> |   GIFT $^{64}_{\overline{O},\overline{D}}$ | 0.010 | 69.1 | 82.3 | 80.4 | 91.9 | 82.2 | 82.3 | 66.9 | 80.6 | 79.5
> | $^B$ GIFT $^{16}_{\underline{QKV},\overline{O},\underline{UG},\overline{D}}$ | 0.201 | **74.6** | **87.9** | **82.3** | **95.6** | 87.1 | **90.3** | **77.9** | **89.0** | **85.6**
>
> **Arithmetic Reasoning (LLaMA-1 13B)**
> |**Method** | **Params (%)** | **AQuA** | **GSM8k** | **MAWPS** | **SVAMP** | **Avg**
> |- | - | - | - | - | - | -
> PrefT | 0.031 | 15.7 | 31.1 | 66.8 | 41.4 | 38.8
> AdapterS | 1.586 | 22.0 | 44.0 | 78.6 | 50.8 | 48.9
> AdapterP | 2.894 | 20.5 | 43.3 | 81.1 | 55.7 | 50.2
> LoRA | 0.67 | 18.5 | 47.5 | 83.6 | 54.6 | 51.1
> GIFT $^{64}_{\underline{Q},\underline{K},\underline{V},\underline{U},\underline{D}}$ | 0.034 | 25.1 | **46.6** | 83.6 | **61.7** | 54.2
> DiReFT | 0.025 | 20.5 | 35.8 | 80.8 | 54.8 | 48.0
> LoReFT | 0.025 | 23.6 | 38.1 | 82.4 | 54.2 | 49.6
> GIFT $^{64}_{\overline{O},\overline{D}}$ | 0.010 | 25.6 | 44.9 | 85.2 | 59.6 | 53.8
> $^B$ GIFT $^{16}_{\underline{QKV},\overline{O},\underline{UG},\overline{D}}$ | 0.201 | **26.0** | 46.2 | **86.3** | 60.6 | **54.8**
>
> - While scalability is an important consideration, we are unable to run experiments on models larger than 13B due to resource limitations as an academic lab. We note that we have submitted the code in the supplementary material to promote follow up work on larger scale experiments.
>
> > Limited Ablation on Layer Selection and Configuration
>
> - In all our experiments, we perform a fair comparison with PEFT methods by fine-tuning the same layers with different methods. We follow [1] (cited as (Hu et al., 2023) in the manuscript) in choosing the layers to finetune in the Commonsense Reasoning and Arithmetic Reasoning tasks (QKVUD), and [2] (cited as (Wu et al. 2024a) in the manuscript) in choosing the layers to finetune for the Instruction Following task. We believe that this is a fair comparison as we are using the same layers as prior works.
> - For a fair comparison with LoReFT and DiReFT, we evaluate $\text{GIFT}_{\overline{O},\overline{D}}$ formulation of GIFT. We beleive these experiments are sufficient to demonstrate the effectiveness of GIFT.
> - We further evaluate a block-wise sharing scheme $^B\text{GIFT}^{16}_{\underline{QKV},\overline{O},\underline{UG},\overline{D}}$, which is unique to GIFT to demonstrate the flexibility of the sharing scheme enabled by GIFT.
>
> **References**
>
> [1] Zhiqiang Hu, Lei Wang, Yihuai Lan, Wanyu Xu, Ee-Peng Lim, Lidong Bing, Xing Xu, Soujanya Poria, Roy Ka-Wei Lee: LLM-Adapters: An Adapter Family for Parameter-Efficient Fine-Tuning of Large Language Models. EMNLP 2023: 5254-5276
>
> [2] Muling Wu, Wenhao Liu, Xiaohua Wang, Tianlong Li, Changze Lv, Zixuan Ling, Jianhao Zhu, Cenyuan Zhang, Xiaoqing Zheng, Xuanjing Huang: Advancing Parameter Efficiency in Fine-tuning via Representation Editing. ACL (1) 2024: 13445-13464

---

> ### Author Response · Authors · 2024-11-21
> **Author Response 2/2**
>
> > How is the comparison result of GIFT among some newly advanced models like gpt4o or others
>
> We would like to clarify that we aim for fair comparisons in experiments following protocols used in the state-of-the-art PEFT/ReFT methods.
> Table 1 is meant to compare with other PEFT/ReFT methods that use the same model (Llama 2 7B) and finetuning dataset (Ultrafeedback). A fair and valid evaluation requires using the same models and datasets as prior methods. As prior methods have demonstrated performance using Llama 2 7B, we have chosen to compare using the same model, and hence we believe that our comparisons are valid and meaningful. We also emphasize that using the same model and training data, none of the prior methods outperform GPT 3.5 Turbo.

---

> ### Author Response · Authors · 2024-11-22
> **Request your feedback and Appreciate your efforts**
>
> Dear Reviewer k2rS,
>
> Hope all is well with you.
>
> We would like to request your feedback on our rebuttal, as well as the revised manuscript at your convenience.
>
> We look forward to it.
>
> Thank you very much.

---

> > ### Author Response · Authors · 2024-11-25
> > **Friendly reminder**
> >
> > Dear Reviewer k2rS,
> >
> > Since the reviewer-author discussion will end tomorrow, we appreciate your initial comments, and would like request and look forward to your comment on our rebuttal and revised submission at your convince.
> >
> > Thank you very much.

---

### Official Review · Reviewer_Rg2D · 2024-11-02

**Soundness:** 2
**Presentation:** 2
**Contribution:** 2
**Rating:** 5
**Confidence:** 3

**Summary:**

This paper proposes a modification to the well-known lora method. Technically, as the original lora method can be formulated as $\overline{W} = W + AB$, the proposed modification changes it to $\overline{W} = W + WAB'$, with $B'$ shared across layers. The authors have discussed the relationship with ReFT, and used experiments to support the efficacy of the modifaction.

**Strengths:**

LoRA is now a widely adopted technique and improvements over it can make profound impacts.

**Weaknesses:**

**Unconvincing method design**

The biggest weakness to me is that the proposed method is not supported by reasonable and convincing motivations. Specifically, there are two major changes from the original LoRA:
1. sharing half of the lora weights across layers;
2. involving the original weight as an extra term into the weight delta.

However, it is unclear why these two changes are useful and how do they work to benefit. After reading the paper I cannot get a satisfying answer. This problem fundamentally limits the value of the work, as it is less likely for people to give the proposed method a try without an intuition that makes people believe it would lead to better results.

**The introduction of the simple method is unnecessarily complicated**

As mentioned in the summary part, the core of the proposed method is simply $\overline{W} = W + AB$ -> $\overline{W} = W + WAB'$, but the paper makes me feel that it is much more complicated.

**Content organization can be better**

The first section is too long. As an introduction section, it involves too many details that are hard to fully understand before reading the methodology section. I suggest only preserving the high-level ideas in this section while moving the technical details elsewhere.

**Others**

1. tied-lora[1] also works on lora + weight-sharing, and I suggest to add some analysis & comparisons.
2. I cannot find the information about the backbone model for visual experiments, welcome to correct me if I missed it.

[1] Renduchintala, Adithya, Tugrul Konuk, and Oleksii Kuchaiev. "Tied-lora: Enhacing parameter efficiency of lora with weight tying." arXiv preprint arXiv:2311.09578 (2023).

**Questions:**

Could authors give some intuitions on why the changes proposed are beneficial?

---

> ### Author Response · Authors · 2024-11-21
> **Author Response 1/2**
>
> Thank you for your valuable feedback. We address your comments as follows:
>
> > Sharing half of the lora weights across layers
>
> - With all due respect, we would like to clarify that there is a key distinction between our proposed approach and your interpretation that our GIFT learns layer specific matrices $A$ and shared matrix $B^\prime$ as $\hat{\omega} = \omega + \omega \cdot A\cdot B^\prime$.
> - This is incorrect: GIFT shares all the learnable parameters across layers. We have revised the writing to make the formulation more clear.
> - GIFT is formulated as $\hat{\omega}^l = \omega^l + \omega^l \cdot \phi \cdot \psi$, where both $\phi$ and $\psi$ are shared across layers (e.g., all query layers selected in fine-tuning).
>
> > Involving the original weight as an extra term into the weight delta
>
> The introduction of the pretrained weights in the delta term is not arbitrary, but a consequence of the generative approach of GIFT in generating the fine-tuned weights from the pretraiend weights. We have revised Section 2 to better formulate this. To  briefly summarize, GIFT aims to learn the fine-tuned weights as $\hat{W}^l_{d_{out}\times d_{in}} = W^l_{d_{out}\times d_{in}} + \mathcal{G}(W^l_{d_{out}\times d_{in}}; \Theta)$. $\mathcal{G}$ generates the residuals as
>
> $\mathcal{G}(\Omega_{L\times d_{out}\times d_{in}};\Theta) = \text{Linear}\Biggl( g\biggl(\text{Linear}\bigl(\Omega_{L\times d_{out}\times d_{in}};\phi\bigr);\theta\biggr);\psi \Biggr)$
>
> where $\Omega_{L\times d_{out}\times d_{in}} = \{W^l_{d_{out} \times d_{in}}\}_{\ \ \forall l \in L}$. As per our hypothesis, $g(\omega^l; \theta)$ can be a simple identity function (verified in Section 4.2 of the revised manuscript). Hence, the formulation of GIFT becomes
>
>  $\hat{\omega}^l = \omega^l_{d_{out}\times d_{in}} + \omega^l_{d_{out}\times d_{in}} \cdot \phi_{d_{in}\times r}\cdot \psi_{r\times d_{in}}$
>
>  > Comparison with Tied LoRA
>
> We have added an ablation study to verify that the generative approach in GIFT with shared learnable parameters is more effective than simply sharing the LoRA weight residuals (i.e., $\Delta W^l = B \cdot A\ \ \forall l \in L$, denoted by Shared LoRA). Table 5 in the revised manuscript shows that GIFT performs much better than Shared LoRA on Commonsense Reasoning using the Llama-3 (8B). This suggests that Tied LoRA, which further ties the residuals for Query, Key and Value together, may not be effective, as it shares the residuals directly across layers (we found it is non-trivial to reproduce Tied LoRA in the HuggingFace's PEFT framework that we use in our experiments, and thus did not directly compare with it in ablation due to time limit). We have reproduced the results of our ablations here for convenience.
>
> **Method** | **Params (%)** | **BoolQ** | **PIQA** | **SIQA** | **HellaS.** | **WinoG.** | **ARC-e** | **ARC-c** | **OBQA** | **Avg**
> | - | - | - | - | - | - | - | - | - | - | -
> Shared LoRA $^{64}_{\underline{Q},\underline{K},\underline{V},\underline{U},\underline{D}}$  | 0.044 | 66.2 | 79.8 | 77.5 | 87.3 | 78.7 | 79.0 | 65.1 | 75.3 | 76.1
> GIFT $^{64}_{\underline{Q},\underline{K},\underline{V},\underline{U},\underline{D}}$ | 0.049 | **75.3** | **89.0** | **81.6** | **96.2** | **88.4** | **92.3** | **81.9** | **87.3** | **86.5**
>
>
> > Content organization can be better
>
> We have organized the content in the revised manuscript to be more precise and concise. We have rephrased the introduction and organized the content to be more straightforward. We have also modified the notation to make it more consistent and clear. Please refer to the revised manuscript and the global comment for the changes made.

---

> ### Author Response · Authors · 2024-11-21
> **Author Response 2/2**
>
> > Benefit of the proposed changes
>
> The proposed changes have several advantages:
> - The layer-wise parameter sharing in GIFT leads to a 14x decrease in the number of trainable parameters compared to LoRA, while outperforming prior PEFT and ReFT methods, as shown through multiple experiments on Commonsense Reasoning, Arithmetic Reasoning, Instruction Following, and Visual Recognition in Section 3.
> - Although VeRA can significantly reduce the number of learnable parameters, the rank of the random matrices needs to be sufficiently high for achieving good performance, which leads to a significant increase in memory consumption and training time in practice (as observed in our experiments). In contrast, GIFT maintains the computational efficiency of LoRA while achieving better performance. For e.g., VeRA takes about 1.5 days in training, while our GIFT takes about 4 hours on the arithmetic reasoning benchmark. The wall time required to train VeRA and GIFT is included in Sections 3.3 and 3.4.
> - The linear formulation of GIFT bridges the gap between PEFT and ReFT (please refer to Section 2.1).
> - The generative approach proposed in GIFT is more holistic in terms of parameter sharing across layers. As opposed to simply sharing the LoRA weight residuals, GIFT generates the residuals in a layer specific way while sharing the trainable parameters across layers. In Section 4.1 of the revised manuscript, we include an ablation study that verifies that the generative approach in GIFT is more effective than simply sharing the LoRA weight residuals.
>
> > Backbone model for visual recognition experiments
>
> We use the ViT-B/16 architecture pretrained on Imagenet21k for the visual experiments. This information is provided in the Models section of the Visual Recognition experiments.

---

> > ### Comment · Reviewer_Rg2D · 2024-11-23
> > **Reply to authors' responses**
> >
> > Thank you for your response. First, I sincerely apologize for the oversights in my initial review and appreciate the authors for pointing them out. I also acknowledge the authors’ efforts in addressing the concerns raised by me and other reviewers in the rebuttal. After reviewing the revised submission, I find significant improvements in clarity and presentation. Based on these considerations, I have updated my score from 3 to 5.
> >
> > However, I still have the following concerns:
> >
> > 1. **Ambiguity in the description of the method**:
> >    My earlier misunderstanding of the method stems from the description in lines 243–246 of the initial submission, which remains in lines 169–171 of the revised version:
> >
> >    > Our GIFT can be treated as the sharing-oriented LoRA, where we have $ B^{l} _ {d_{\text{out}} \times r} = \omega^{l} _ {d _ {\text{out}} \times d _ {\text{in}}} \phi _ {d_{\text{in}} \times r}$,  conditioned on the pretrained weights and retained to be **layer-specific**, and $ A^{l} _ {r \times d_{\text{in}}} = \psi _ {r \times d_{\text{in}}} (\forall l) $ relaxed to be **layer-agnostic**.
> >
> >    This wording led me to believe that $ \phi$ is layer-specific while $ \psi $ is shared across layers. Although I now understand the intended meaning, I still think the description can easily lead to misinterpretation. I strongly suggest revising it.
> >
> > 2. **Lack of convincing explanation for the effectiveness of GIFT**:
> >    I remain unconvinced by the authors’ response to my question about **why** the proposed changes are beneficial. Specifically, I am seeking an explanation for **why** GIFT can achieve better performance than other methods with fewer trainable parameters and lower training costs. The authors’ response primarily reiterates three points:
> >    - GIFT uses fewer trainable parameters and requires lower training costs (which is already well illustrated in the paper).
> >    - GIFT demonstrates better results on benchmarks.
> >    - GIFT conceptually unifies PEFT and ReFT. (While I agree that such conceptual elegance is valuable, it does not inherently explain the improvement in performance.)
> >
> >    Let me clarify my concern further: Typically, improving the accuracy of deep learning methods requires more trainable parameters and higher training costs. In contrast, the authors claim to have proposed a method that achieves higher accuracy with fewer trainable parameters and lower training costs, this means that GIFT should have leveraged some inductive bias in its design that makes it better suited to the characteristics of the target tasks and data. However, the paper lacks an analysis or discussion on what inductive biases GIFT exploits and how these biases align with the properties of the tasks or data.
> >
> >    **Why this matters**: LoRA is a widely successful and recognized method across various scales, modalities, and tasks. While the evaluation of GIFT in this paper is academically thorough, it is far from the extensive real-world testing that LoRA has undergone. To convincingly demonstrate that GIFT is superior to LoRA, it is not enough to show experimental results on certain benchmarks. There must also be a theoretical justification (high-level intuitive explanations are enough and I don't mean I need bunches of formulas) explaining why GIFT might have a better inductive bias than other PEFT methods. Without analyzing the causes of its performance, it will also be harder for future work to build on the success of GIFT.
> >
> > Based on the above considerations, I still do not recommend acceptance of this paper at this time. However, I am open to acceptance under one of the following conditions:
> > 1. The authors provide a more compelling explanation within the remaining rebuttal period.
> > 2. Other reviewers and the AC believe that the experimental results are sufficient to compensate for the paper’s other issues.
> > 3. Other reviewers and the AC consider the authors’ current response to be already sufficiently convincing.

---

> > > ### Author Response · Authors · 2024-11-23
> > > **Thank you for your response**
> > >
> > > Dear Reviewer Rg2D,
> > >
> > > Thank you very much for your response. We appreciate your efforts in helping us improve the quality of our submission.
> > >
> > > 1. **Ambiguity in the description of the method**:
> > >
> > > Sorry for not being thoroughly clear in the wording. We further revise the the wording as follows and will update the manuscript:
> > >
> > > > ... where we have the counterpart of the layer-specific $B^l_{d_{out}\times r}$ in LoRA, $B^l_{d_{out}\times r}=W^l_{d_{out}\times d_{in}} \cdot  \phi_{d_{in}\times r}$, is computed, rather than being treated as direct learnable parameters, by  conditioning on the layer-specific pretrained weights and modulating with a layer-agnostic $\phi_{d_{in}\times r}$, and  the counterpart of the layer-specific $A^l_{r\times d_{in}}$ in LoRA, $A^l_{r\times d_{in}}=\psi_{r\times d_{in}} (\forall l)$ is directly relaxed to be layer-agnostic.
> > >
> > > 2. **Lack of convincing explanation for the effectiveness of GIFT**:
> > >
> > > We totally agree with you that we should do our best in understanding why a method such as our GIFT works better than baselines. We also totally agree that LoRA has been much thoroughly tested with the efforts from the entire community of PEFT.  We appreciate your understanding of our experiments on GIFT are academically thorough.
> > >
> > > Let's try to address your remaining concern.
> > >
> > > > Typically, improving the accuracy of deep learning methods requires more trainable parameters and higher training costs. In contrast, the authors claim to have proposed a method that achieves higher accuracy with fewer trainable parameters and lower training costs, this means that GIFT should have leveraged some inductive bias in its design that makes it better suited to the characteristics of the target tasks and data. However, the paper lacks an analysis or discussion on what inductive biases GIFT exploits and how these biases align with the properties of the tasks or data.
> > >
> > > >> In our understanding, it is generally true for training neural networks **from scratch** that "improving the accuracy of deep learning methods requires more trainable parameters and higher training costs", and those neural networks in comparison are of similar types, since we have witnessed many progress in the literature smaller and/or more efficient neural networks can outperform larger ones, e.g., MobileNets and EfficientNets in computer vision tasks.
> > >
> > > >> In this paper, we focus on efficiently fine-tuning pretrained Transformer backbones.   The pretrained Transformer backbones themselves are the main source of inductive biases to be leveraged.  We try to explain this in Section 2.3 in the revised manuscript, which we reproduce below for your information:
> > >
> > > >> **2.3 GIFT Aims to ``Balance'' Pretraining and Fine-Tuning**
> > >
> > > >> Pretrained Transformer backbones encode diverse knowledge from large-scale pretraining datasets within their weights. Fine-tuning them for a downstream task aims to incorporate new information from the task-specific training data and utilize the information present in the pretrained weights to the fullest extent. To achieve this, the fine-tuned weights can be directly conditioned on the pretrained weights, such that the new information is learned conditionally from the information in the pretrained weights. While LoRA and it's variants use a residual structure to address this, the residual weights are not directly conditioned on the pretrained weights, but rather learned via back-propagation (chain rule) updates. One of the simplest functions that can achieve this explicit conditioning is a linear transformation of the pretrained weights, as leveraged in Eqn.7. Hence, the fine-tuned weights can also be expressed in the space of the pretrained weights $W_{d_{out}\times d_{in}}$ via $W_{d_{out}\times d_{in}}\cdot \Theta_{d_{in}\times d_{in}}$.
> > >
> > > >> When pretrained Transformer backbones are sufficiently expressive, as is typically assumed in efficient fine-tuning, simpler parameterization methods like GIFT should be more generalizable and better under the principle of Occam's razor. Our ablation studies in Section 4.2 show the effectiveness of the linear parametrization over other schemes.
> > >
> > > >> We also compare the gradient updates between LoRA and our GIFT using an exampling of fine-tuning a toy MLP in the Appendix B, which shows the layer-agnostic parameters ($\phi$ and $\psi$) in our GIFT gather information from all the layers, which might be potential reasons why they can be effectively learned.
> > >
> > > Please kindly inform us what we can do further to potentially clarify your concerns. We really appreciate your valuable feedback again.

---

> > > > ### Comment · Reviewer_Rg2D · 2024-11-24
> > > > **Following response**
> > > >
> > > > To me, the statement, *"While LoRA and its variants use a residual structure to address this, the residual weights are not directly conditioned on the pretrained weights, but rather learned via back-propagation (chain rule) updates,"* is the only explanation provided that aligns with the way of reasoning that I expect for explaining why GIFT is superior to methods like LoRA. However, such a claim without empirical evidence is far from convincing.
> > > >
> > > > Let me give an example of what I believe is necessary:
> > > >
> > > > The modeling process of GIFT appears to be as follows (hopefully I have no more misunderstandings):
> > > > - **Step 0**: Start with the LoRA framework.
> > > > - **Step 1**: Incorporate the original weight term into the multiplication (changing the delta formulation from \( \Delta = AB \) to \( \Delta = ABW \)).
> > > > - **Step 2**: Share the matrices \( A \) and \( B \) across layers.
> > > >
> > > > When you claim that *"the problem of LoRA is that the delta is not explicitly conditioned on \( W \), but only implicitly through backpropagation,"* you are equivalently claiming that **Step 1**—explicitly conditioning the delta on \( W \)—is beneficial for PEFT. If so, you need experiments to demonstrate the impact of simply adding the \( W \) term to LoRA. Would it improve the performance? if yes, how significant/robust? If not, why, and does that mean weight sharing is the key element for good performance? ....
> > > >
> > > > Then you can move forward to **Step 2**. What its role? While it clearly reduces the number of trainable parameters, is this operation lossless in terms of accuracy? If it is lossless, why? Are there specific factors that make the method after **Step 1** more resilient to weight sharing? If it is lossy, how significant is the loss in performance, and why you involve this to your methodology design?
> > > >
> > > > Again, I don't mean the paper has to be organized in such a way. It's just an example.
> > > >
> > > > In conclusion, I hope the methodology can be developed based on motivation that is supported by relatively strict deductions and validations. However, the current explanations feel like castles in the air, which personally makes me reluctant to recommend acceptance.

---

> ### Author Response · Authors · 2024-11-22
> **Request your feedback and Appreciate your efforts**
>
> Dear Reviewer Rg2D,
>
> Hope all is well with you.
>
> We would like to request your feedback on our rebuttal, as well as the revised manuscript at your convenience.
>
> We look forward to it.
>
> Thank you very much.

---

> ### Author Response · Authors · 2024-11-24
> **Thank you for your further questions**
>
> Dear Reviewer Rg2D,
>
> We appreciate your valuable time and efforts.
>
> We hope we have addressed your concerns in the initial review and your follow-up questions on the inductive biases.   **We also respect your decision.**
>
> We would like address your latest concerns as follows.
>
> > The modeling process of GIFT appears to be as follows (hopefully I have no more misunderstandings):
>
> >> Every submission may have an exploration journey under the hood.  We would like to share some of our efforts to clarify things.  When we developed our GIFT, we start with computer vision tasks, and with more sophisticated realization of the $g(\cdot; \Theta)$ in Eqn.6. We show the ablations studies in Sec.4.2 and Table 6.  We obtained very promising performance, but then challenged ourselves to seek simpler formulation, resulting in the simple GIFT in this submission.  After we observed that  the simple GIFT works well, we start to explore its relationships to other PEFT and later on ReFT.
>
> >> **It is our strong believe that finding a simple formulation that works well in practice worths revealing to the community.**  Of course, we all have future work to do to make our methods better and stronger.
>
> >>  **We would like to point out a typo in your suggestion**: In LoRA, we have \Delta=BA, not \Delta =AB; Following your specification of \Delta=AB  to  \Delta = ABW, which does not hold, since $B\in \mathbb{R}^{r\times d_{in}}$, $A\in \mathbb{R}^{d_{out}\times r}$, and $W\in \mathbb{R}^{d_{out}\times d_{in}}$.
>
> >> If we understand your intent correctly, we have applied our GIFT to the $d_{dout}$ dimension too (Eqn.8), and propose a block-wise sharing configuration:  $^{B}$ GIFT $^{r}_{\underline{QKV},\overline{O},\underline{UG},\overline{D}}$ ,
>
> >> which consists of GIFTs applied to both $d_{in}$ (underline) and $d_{out}$ (upperline) dimensions, and shows stronger consistency of achieving better results across tasks (Commonsense Reasoning and Arithmetic Reasoning), as we highlighted in the global response.

---

### Author Response · Authors · 2024-11-21
**Brief summary of Results and Changes**

We thank the reviewers for their valuable feedback. We have carefully revised the original manuscript based on the comments provided. Below, we briefly summarize our results, followed by an overview of the changes made in the revised manuscript. We believe that the revised manuscript is clearer and that the changes implemented in response to the reviewers' suggestions have significantly enhanced the overall quality of the paper. We have addressed the concerns raised by reviewers through individual comments.

# Brief summary of results
- **Instruction Following** (Section 3.2), our GIFT can outperform GPT-3.5 Turbo using 0.0311% trainable parameters in fine-tuning Llama-2 (7B), which is the only method to do so in our comparisons.
- **Commonsense Reasoning** (Section 3.3), our GIFT outperforms both the prior art of PEFT and of ReFT consistently using Llama 1-2-3 model family, often by large margin with less trainable parameters used.
- **Arithmetic Reasoning** (Section 3.4), our GIFT can outperform all the prior PEFT and ReFT approaches. Unlike VeRA, which performs slightly better than LoRA, GIFT maintains the computational efficiency while achieving better performance. VeRA takes about 1.5 days in training, while our GIFT takes about 4 hours.
- The proposed block-wise sharing $^{B}$ GIFT $^{r}_{\underline{QKV},\overline{O},\underline{UG},\overline{D}}$ shows stronger consistency of achieving better results across tasks (Commonsense Reasoning and Arithmetic Reasoning).

# Summary of changes
As suggested by reviewers `Rg2D` and `53Qz`, we have rephrased the introduction and organized the content to more precise and concise. Following are the changes made in the revised manuscript:

## Section 1: Introduction
- Section 1 now illustrates the motivation, formulation, contributions and design choices in a more straightforward manner.
- As suggested by reviewer `53Qz`, we have modified the notation to make it more consistent and clear.

## Section 2: Approach
We have removed the redundancies in Section 2, which now focusses on the generic formulation for the generative approach proposed in GIFT, highlighting the key properties of the method:
- GIFT shares all the learnable parameters across layers (for e.g., one GIFT is shared across all Query layers), but still generates layer specific finetuning residuals and fine-tuned weights because of the generative nature.
- GIFT bridges PEFT and ReFT, as it can be equivalently applied to the representations or the weights, and is applied uniformly to all the tokens in the sequence, eliminating the need for a hyperparameter search over the token selection.

## Section 3: Experiments
- As suggested by reviewer `k2rS`, we have added experiments with Llama-1 13B for Commonsense Reasoning and Arithmetic Reasoning tasks. Table 2 and Table 3 in the revised manuscript show that GIFT performs better/on par with prior PEFT and ReFT methods while being significantly more parameter efficient. This shows that GIFT can be scaled to larger models on the scale of 13B, and can potentially be scaled to even larger models. Due to resource constraints, we are unable to run experiments on larger models on the scale of 65B.
- We include additional experiments with VeRA for Commonsense Reasoning, and show that GIFT outperforms VeRA by a significant margin, while requiring less wall time to train. We have also added the wall time required to train VeRA and GIFT in Sections 3.2 and 3.3.

## Section 4: Albation Studies
We have organized the ablation studes in a separate section, which now includes an additional ablation study that verifies that the generative approach in GIFT with shared learnable parameters is more effective than simply sharing the LoRA weight residuals (i.e., $\Delta W^l = B \cdot A\ \ \forall l \in L$, denoted by Shared LoRA).
- Table 5 in the revised manuscript shows that GIFT performs much better than Shared LoRA on Commonsense Reasoning using the Llama-3 (8B). This suggests that Tied LoRA, suggested by reviewer `Rg2D` may not be effective, as it shares the residuals directly across layers.

---

### Author Response · Authors · 2024-12-01
**Summary of author-reviewer discussion and request reviewers' (re-)evaluation of the changes in revision**

Dear Reviewers,

We are grateful for your efforts and valuable time spent on our submission and revision, which has helped us a lot in improving the quality of our submission.

We try to do our diligence to maximize the outcome from this valuable reviewing process. So, we briefly summarize the current status as follows:

Reviewer `k2rS` has yet to give their feedback on our rebuttal. We hope to have their valuable feedback soon and appreciate the time spent on our submission.

Reviewers `Rg2D` and `53Qz` have some further concerns as follows:

> Lack of explanation for the effectiveness of GIFT

We have tried to address reviewer `Rg2D`'s concern in Section 2.3 of the revised manuscript, which we have reproduced below:

>> Pretrained Transformer backbones encode diverse knowledge from large-scale pretraining datasets within their weights. Fine-tuning them for a downstream task aims to incorporate new information from the task-specific training data and utilize the information present in the pretrained weights to the fullest extent. To achieve this, the fine-tuned weights can be directly conditioned on the pretrained weights, such that the new information is learned conditionally from the information in the pretrained weights. While LoRA and it's variants use a residual structure to address this, the residual weights are not directly conditioned on the pretrained weights, but rather learned via back-propagation (chain rule) updates. One of the simplest functions that can achieve this explicit conditioning is a linear transformation of the pretrained weights, as leveraged in Eqn.7. Hence, the fine-tuned weights can also be expressed in the space of the pretrained weights $W_{d_{out}\times d_{in}}$ via $W_{d_{out}\times d_{in}}\cdot \Theta_{d_{in}\times d_{in}}$.

We agree that a more thorough explanation of the effectiveness of GIFT would be beneficial. We also strongly believe that finding a simple formulation that works well in practice worths revealing to the community, and plan to work on a more detailed explanation (such as the explorations and analyses on learning rates for $\phi$ and $\psi$ as done for $B$ and $A$ in LoRA by [1] and the gradient decomposition based initialization for $\phi$ and $\psi$ as done for $B$ and $A$ in LoRA by [2]) in future work.

> Concerns about the notation used in GIFT

As suggested by reviewer `53Qz`, we have modified the notation slightly by using $W$ instead of $\omega$ to represent the pretrained weights. However, we retain the use of $\phi$ and $\psi$ to denote the learnable parameters, as they are a part of the parameters of the weight-generator network. This lets us distinguish between standard fine-tuning methods and GIFT. Our GIFT generates the fine-tuned weights from the pre-trained weights, rather than learning them as model parameters. This also keeps the notation consistent across equations (4), (5) and (6), which are,

$\hat{W}^l_{d_{out}\times d_{in}} = W^l_{d_{out}\times d_{in}} + \mathcal{G}(W^l_{d_{out}\times d_{in}}; \Theta)$ (.... Eq. 4)

$\qquad \qquad =W^l_{d_{out}\times d_{in}} +$ $W^l_{d_{out}\times d_{in}} \cdot \phi_{d_{in}\times r}\cdot \psi_{r\times d_{in}}$ (.... Eq. 5)

$\mathcal{G}(\Omega_{L\times d_{out}\times d_{in}};\Theta) = \text{Linear}\Biggl( g\biggl(\text{Linear}\bigl(\Omega_{L\times d_{out}\times d_{in}};\phi\bigr);\theta\biggr);\psi \Biggr)$ (.... Eq. 6)

Overall, we believe the notations used in our GIFT is self-contained and clear.

Thank you very much.

---
[1] S. Hayou, N. Ghosh and B. Yu, "The Impact of Initialization on LoRA Finetuning Dynamics", http://arxiv.org/abs/2406.08447

[2] S. Wang, L. Yu and Jianli, "LoRA-GA: Low-Rank Adaptation with Gradient Approximation", http://arxiv.org/abs/2407.05000

---

### Meta-Review · Area_Chair_QCLP · 2024-12-20

**Metareview:**

The paper introduces a method for parameter-efficient fine-tuning that aims to improve model performance while reducing the number of trainable parameters. The reviewers appreciated the strong experimental results demonstrating its advantage over baseline methods like lora.

However, the reviewers expressed concerns about the unclear motivation for the proposed modifications to lora, particularly the weight-sharing and inclusion of original weights in the delta term, suggesting a more intuitive explanation is needed. The methodology was also seen as overly complicated for a simple update, and the paper's organization could be improved, particularly in the introduction. Additionally, the paper lacked sufficient comparisons with full fine-tuning, especially in commonsense and arithmetic reasoning tasks. There were no ablation studies on layer selection. Some of these concerns are addressed and confirmed by reviewers, while some of them are not well addressed. Some of these concerns are addressed and confirmed by reviewers, while some of them are not well addressed.

Considering the average score of 5, which is below the acceptance threshold, and the overall feedback, the final decision for the paper is reject.

**Additional Comments On Reviewer Discussion:**

The reviewers expressed concerns about the unclear motivation for the proposed modifications to lora, particularly the weight-sharing and inclusion of original weights in the delta term, suggesting a more intuitive explanation is needed. The methodology was also seen as overly complicated for a simple update, and the paper's organization could be improved, particularly in the introduction. Additionally, the paper lacked sufficient comparisons with full fine-tuning, especially in commonsense and arithmetic reasoning tasks. There were no ablation studies on layer selection. Some of these concerns are addressed and confirmed by reviewers, while some of them are not well addressed. Some of these concerns are addressed and confirmed by reviewers, while some of them are not well addressed.

---

### Decision · Program_Chairs · 2025-01-22

Reject